# Evolution of reduced co-activator dependence led to target expansion of a starvation response pathway

**Bin Z He[1]\*, Xu Zhou[1]†, Erin K O'Shea[1,2,3]\***

[1]Faculty of Arts and Sciences Center for Systems Biology, Howard Hughes Medical Institute, Harvard University, Cambridge, United States; [2]Department of Molecular and Cellular Biology, Harvard University, Cambridge, United States; [3]Department of Chemistry and Chemical Biology, Harvard University, Cambridge, United States

**Abstract** Although combinatorial regulation is a common feature in gene regulatory networks, how it evolves and affects network structure and function is not well understood. In *S. cerevisiae*, the phosphate starvation (PHO) responsive transcription factors Pho4 and Pho2 are required for gene induction and survival during phosphate starvation. In the related human commensal *C. glabrata*, Pho4 is required but Pho2 is dispensable for survival in phosphate starvation and is only partially required for inducing PHO genes. Phylogenetic survey suggests that reduced dependence on Pho2 evolved in *C. glabrata* and closely related species. In *S. cerevisiae*, less Pho2-dependent Pho4 orthologs induce more genes. In *C. glabrata*, its Pho4 binds to more locations and induces three times as many genes as Pho4 in *S. cerevisiae* does. Our work shows how evolution of combinatorial regulation allows for rapid expansion of a gene regulatory network's targets, possibly extending its physiological functions.

**\*For correspondence:**
binhe@fas.harvard.edu;
empty.hb@gmail.com (BZH);
osheae@hhmi.org (EKO'S)

**Present address:** †Yale School of Medicine, New Haven, United States

## Introduction

Evolution of gene regulatory networks (GRNs) is a major source of phenotypic diversity (*Wray, 2007*; *Stern and Frankel, 2013*; *Prud'homme et al., 2006*; *Gompel et al., 2005*; *Jones et al., 2012*; *Wang et al., 1999*). One common feature of GRNs is combinatorial regulation by multiple transcription factors (TFs) – for example, the co-regulation of circadian gene expression in cyanobacteria by both the cell-autonomous clock and the external conditions (*Espinosa et al., 2015*), and the determination of cell fates by multiple 'selector genes' in animal development (*Mann and Carroll, 2002*). Not only is combinatorial regulation important for GRN function, it also contributes to its evolution through changes in protein-protein or protein-DNA interactions (*Kirschner and Gerhart, 2006*, chap. 4; *Tsong et al., 2003*; *Baker et al., 2012*; *Brayer et al., 2011*). The consequences of such changes can be either conserved network output (*Tsong et al., 2006*) or evolution of new network function (*Tuch et al., 2008*).

Despite a rich literature on GRN evolution, few studies have documented the evolution of combinatorial regulation and its influence on network structure and function (*Tuch et al., 2008*; *Baker et al., 2012*). Moreover, the existing literature on GRN evolution is strongly biased towards developmental networks (*Stern, 2010*; *Peter and Davidson, 2011*). While such networks provide attractive attributes, such as visible phenotypes and well-resolved genetic underpinning, it has been suggested that network architecture strongly influences the tempo and mode of its evolution (*Erwin and Davidson, 2009*; *Wittkopp, 2007*). Therefore, it is unclear whether all GRNs follow similar or different rules during their evolution.

To approach this question we studied the regulatory divergence in the phosphate starvation (PHO) response network in yeast. For three reasons, this system is well suited for our question. First,

**eLife digest** The diversity of life on Earth has intrigued generations of scientists and nature lovers alike. Research over recent decades has revealed that much of the diversity we can see did not require the invention of new genes. Instead, living forms diversified mostly by using old genes in new ways – for example, by changing when or where an existing gene became active. This kind of change is referred to as "regulatory evolution".

A class of proteins called transcription factors are hot spots in regulatory evolution. These proteins recognize specific sequences of DNA to control the activity of other genes, and so represent the "readers" of the genetic information. Small changes to how a transcription factor is regulated, or the genes it targets, can lead to dramatic changes in an organism. Before we can understand how life on Earth evolved to be so diverse, scientists must first answer how transcription factors evolve and what consequences this has on their target genes.

So far, most studies of regulatory evolution have focused on networks of transcription factors and genes that control how an organism develops. He et al. have now studied a regulatory network that is behind a different process, namely how an organism responds to stress or starvation. These two types of regulatory networks are structured differently and work in different ways. These differences made He et al. wonder if the networks evolved differently too.

The chemical phosphate is an essential nutrient for all living things, and He et al. compared how two different species of yeast responded to a lack of phosphate. The key difference was how much a major transcription factor known as Pho4 depended on a so-called co-activator protein named Pho2 to carry out its role. Baker's yeast (*Saccharomyces cerevisiae*), which is commonly used in laboratory experiments, requires both Pho4 and Pho2 to activate about 20 genes when inorganic phosphate is not available in its environment. However, in a related yeast species called *Candida glabrata*, Pho4 has evolved to depend less on Pho2. He et al. went on to show that, as well as being less dependent on Pho2, Pho4 in *C. glabrata* activates more than three times as many genes as Pho4 in *S. cerevisiae* does in the absence of phosphate. These additional gene targets for Pho4 in *C. glabrata* are predicted to extend the network's activities, and allow it to regulate new process including the yeast's responses to other types of stress and the building of the yeast's cell wall.

Together these findings show a new way that regulatory networks can evolve, that is, by reducing its dependence on the co-activator, a transcription factor can expand the number of genes it targets. This has not been seen for regulatory networks related to development, suggesting that different networks can indeed evolve in different ways. Lastly, because disease-causing microbes are often stressed inside their hosts and *C. glabrata* sometimes infects humans, understanding how this yeast's response to stress has evolved may lead to new ways to prevent and treat this infection.

starvation/stress response networks differ in architecture from developmental networks, leading us to expect differences in their evolutionary patterns. Second, the GRN controlling the PHO response has been well studied in the model yeast *S. cerevisiae*, providing a solid foundation for comparative analyses. Third, the two species we focus on, *S. cerevisiae* and a human commensal and opportunistic pathogen *C. glabrata*, occupy distinct ecological niches but share ~90% of their gene repertoire and have an average of ~67% protein sequence identity (*Gabaldón et al., 2013*), enabling us to trace the evolution of the network by studying the function of orthologous regulatory proteins.

Phosphate is an essential nutrient for all organisms. To maintain phosphate homeostasis, *S. cerevisiae* activates a phosphate starvation pathway in response to limitation for inorganic phosphate (*Ogawa et al., 2000*). In phosphate replete conditions, the transcription factor Pho4 is phosphorylated and localized to the cytoplasm, and phosphate response genes (PHO genes) are not expressed (*O'Neill et al., 1996*). As the concentration of extracellular inorganic phosphate (Pi) drops, cells activate the phosphate starvation response and the dephosphorylated Pho4 is imported into the nucleus, where it functions together with the homeodomain transcription factor Pho2 to activate PHO gene expression (*O'Neill et al., 1996*; *Vogel et al., 1989*; *Barbarić et al., 1996*; *Barbaric et al., 1998*; *Shao et al., 1996*).

Although Pho4 binds to ~100 locations in the *S. cerevisiae* genome, it regulates fewer than 30 genes (*Zhou et al., 2011*). Only genes at which Pho2 and Pho4 bind cooperatively in the promoter

region are activated, indicating that Pho2 increases the selectivity of the gene set induced in response to phosphate starvation (*Zhou et al., 2011*). In *C. glabrata*, Pho4 and Pho2 orthologs (hereinafter referred to as CgPho4 and CgPho2) exist, but unlike Pho4 and Pho2 in *S. cerevisiae* (hereinafter referred to as ScPho4 and ScPho2), CgPho4 can induce gene expression in the absence of CgPho2 (*Kerwin and Wykoff, 2009*). This change in the dependence on the co-activator is not due to a higher expression level of CgPho4 or changes in the promoter regions of its target genes, and therefore is likely the result of alterations in the function of CgPho4 (*Kerwin and Wykoff, 2009*).

We investigated the evolution of the PHO pathway in a diverse group of yeast species known as Hemiascomycetes (*Knop, 2006*; *Diezmann et al., 2004*), which includes but is not limited to *S. cerevisiae*, *C. glabrata*, *K. lactis*, *C. albicans* and *Y. lypolitica*, and found that *PHO4* and *PHO2* are conserved as single copy genes in this group. We first evaluated Pho4 orthologs from a representative set of this group of species for their ability to activate gene expression in the absence of Pho2 in the *S. cerevisiae* background – this allowed us to establish that the reduced dependence on Pho2 evolved in a species clade that includes *C. glabrata*, two other human commensal yeasts and an environmental species. We then used functional genomics to assess the consequence of reduced Pho2 dependence on gene expression. Finally, we identified the *bona fide* targets of Pho4 in *C. glabrata* and compared them to Pho4 targets in *S. cerevisiae*. Our results show that evolution of combinatorial regulation, in terms of dependence of the major transcription factor on the co-activator, contributes significantly to the evolution of a gene regulatory network's targets and functions, and as a result may lead to a new physiological response to the stress.

## Results

### Evolution of Pho2-dependence among Pho4 orthologs in Hemiascomycetes

We first confirmed the previously reported result that deleting *PHO2* in *C. glabrata* does not eliminate expression of the secreted phosphatase encoded by *PMU2* (*Orkwis et al., 2010*) (*Figure 1A*), and then demonstrated that, in contrast to ScPho2, CgPho2 is dispensable for survival in phosphate-limited conditions (*Figure 1B*). It has been shown that CgPho4 is able to induce phosphatase expression in *S. cerevisiae* without ScPho2, strongly suggesting that changes in Pho4 are primarily responsible for the difference in Pho2-dependence (*Kerwin and Wykoff, 2009*). To understand whether dependence on Pho2 in *S. cerevisiae* is the ancestral or the derived state and how this property of Pho4 evolved among related species, we surveyed the activity of Pho4 orthologs from 16 species in the Hemiascomycete class. To isolate the changes in Pho4 activity from the genomic background (e.g. promoter and Pho2 changes), we inserted coding sequences (CDSs) of the 16 Pho4 orthologs into an *S. cerevisiae* background lacking both the endogenous ScPho4 CDS and the negative regulator of the PHO pathway, Pho80, under the control of the endogenous ScPho4 promoter. Deletion of *PHO80*, the cyclin component of the cyclin-dependent-kinase complex, causes constitutive nuclear localization of Pho4 and de-repression of PHO genes, mimicking phosphate starvation (*O'Neill et al., 1996*; *Huang et al., 2005*). The use of the *pho80Δ* strain background allowed us to decouple the gene induction ability of Pho4 orthologs with or without ScPho2 from viability (below).

To determine the level of dependence on Pho2 for each Pho4 ortholog, we compared its activity, reflected by induction of the secreted phosphatase encoded by *PHO5*, in the presence or absence of ScPho2. We first evaluated whether the Pho4 orthologs can functionally compensate for ScPho4 in the *S. cerevisiae* background lacking the Pho4 negative regulator Pho80 and paired with ScPho2 (*Figure 2A*, left three columns). We found that the majority of Pho4 orthologs were able to induce Pho5 to a level significantly above the background (*pho4Δ*, bottom row in *Figure 2A*), although the activity declines with increasing evolutionary distance from *S. cerevisiae*. When we measured the activity of the Pho4 orthologs in the absence of Pho2 (*Figure 2A*, right three columns), we found that Pho4 orthologs from the clade consisting of *C. glabrata*, *C. bracarensis*, *N. delphensis* and *C. nivariensis* (hereafter the 'glabrata clade') were able to induce Pho5 expression in the absence of Pho2 (*Figure 2A,B*), but Pho4 orthologs from outgroup species such as *N. castellii*, *K. lactis*, and *L. waltii* could not. We conclude that the common ancestor of *S. cerevisiae* and *C. glabrata* had a Pho4

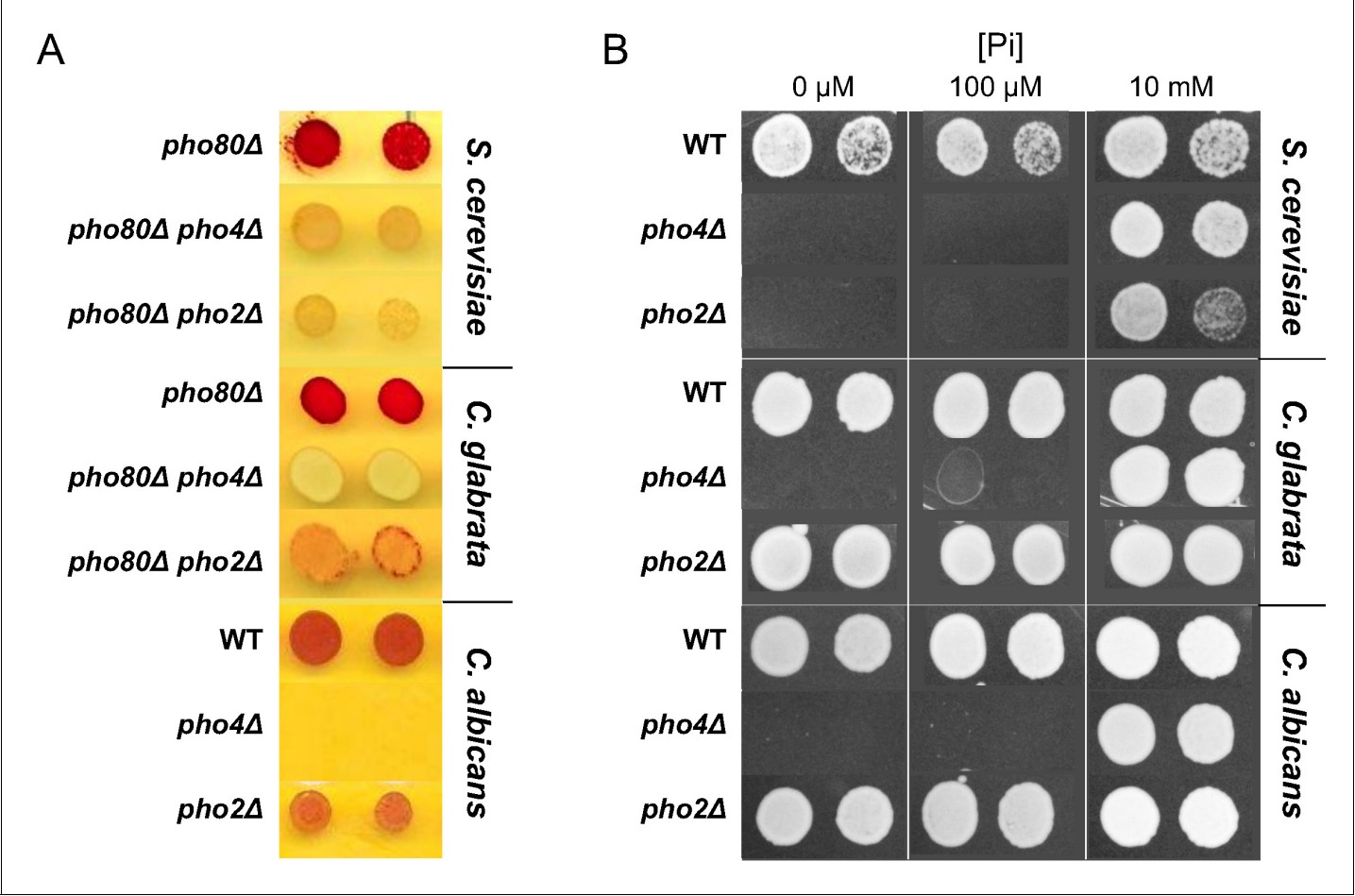

**Figure 1.** Difference among three yeast species in their dependence on Pho2 for gene induction and organismal survival under low Pi conditions. (A) Induction of the secreted phosphatase in each species measured by a semi-quantitative acid phosphatase assay (*Wykoff et al., 2007*). The intensity of the red color indicates the total activity of the secreted acid phosphatase from the cell colony. For *S. cerevisiae* and *C. glabrata*, strains lacking the negative regulator of Pho4 – Pho80 – were spotted on synthetic medium with 10 mM Pi. For *C. albicans*, strains with *PHO80* wild-type were spotted on synthetic medium lacking inorganic phosphate; the *pho2Δ* strain was not able to grow on this plate. (B) Colony growth phenotype of the wild-type, *pho4Δ*, *pho2Δ* strains in each of the three species, under different Pi concentrations. In both panels, two technical replicates of four-fold serial dilutions from the same culture are shown for each strain.

that is dependent on Pho2 and that the reduced dependence was evolutionarily derived in the *glabrata* clade.

Surprisingly, the Pho4 ortholog from a distantly related commensal and pathogenic yeast *C. albicans* (hereinafter 'Ca') weakly induced Pho5 in a Pho2-independent manner (*Figure 2A*). We further demonstrated that *C. albicans* does not require Pho2 for survival in phosphate-limited conditions, and deletion of Pho2 does not abolish the induction of the secreted phosphatase in that species (*Figure 1*). We were not able to infer whether the reduced Pho2 dependence in *C. albicans* represents the derived or the ancestral state, because the Pho4 ortholog from *Y. lipolytica*, an outgroup of *C. albicans* and *S. cerevisiae*, failed to complement ScPho4 in *S. cerevisiae*.

In total, we identified five Pho4 orthologs with reduced dependence on Pho2. Notably, the extent of the reduction varies between the *glabrata* clade Pho4 orthologs, suggesting that the strength of combinatorial regulation is a quantitative trait that can be fine-tuned by mutations during evolution.

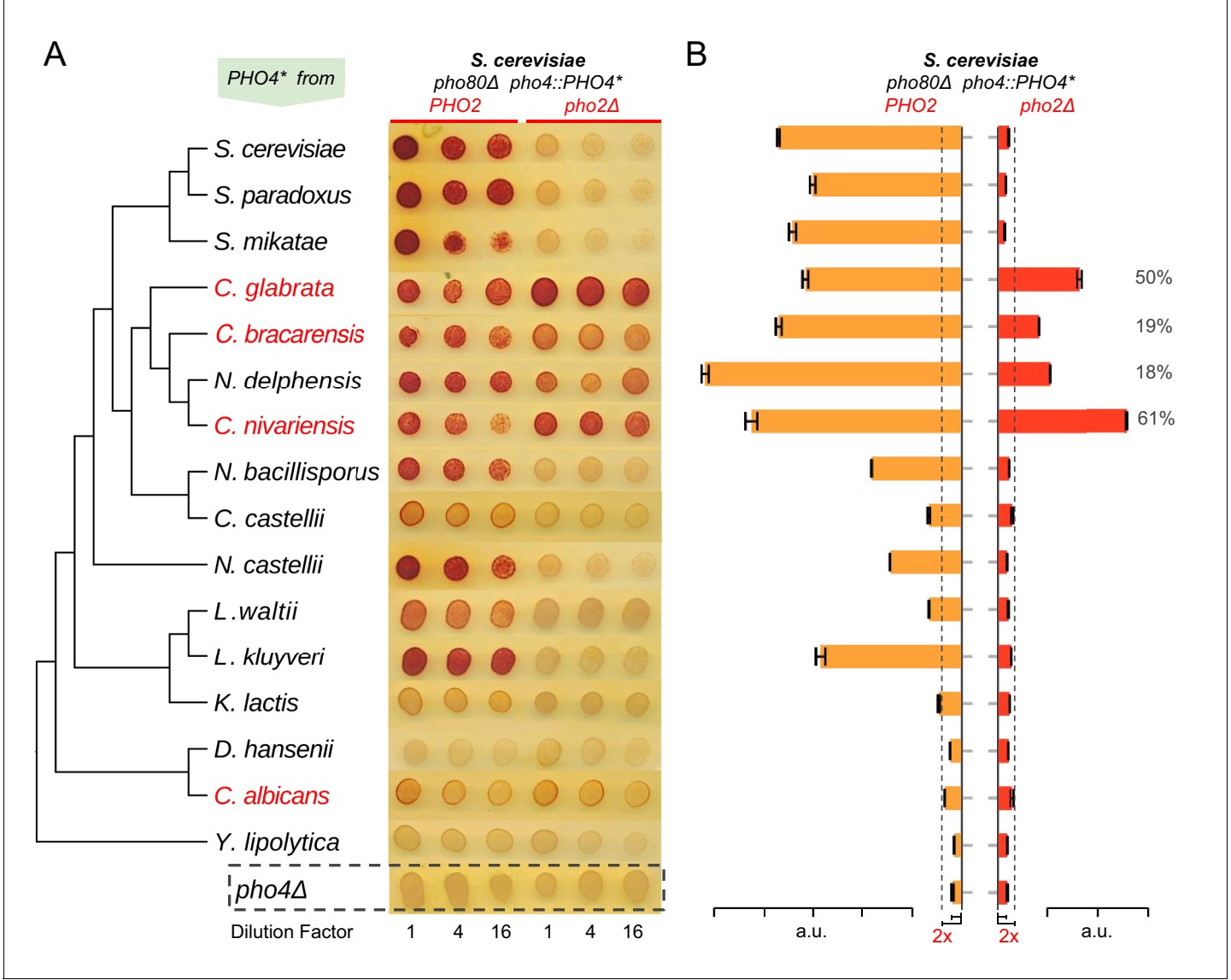

**Figure 2.** Evolution of Pho4 dependence on Pho2 in the Hemiascomycetes. (**A**) Survey of Pho4 orthologs activity in the *S. cerevisiae* background by the semi-quantitative acid phosphatase assay with or without ScPho2. The species phylogenetic relationship shown on the left were based on (*Wapinski et al., 2007*). Species names marked in red indicate known commensal and human pathogens. All strains were constructed in an identical *S. cerevisiae* background lacking the PHO pathway negative regulator Pho80. For each strain, three technical replicates in four-fold serial dilutions were assayed. A strain lacking Pho4 serves as the negative control (*pho4Δ*, dotted box). (**B**) Quantitative phosphatase assay for the same strains in (**A**). The bar graph shows the mean and standard deviation of the secreted phosphatase activity (N = 2, technical replicates). For Pho4 orthologs with noticeable activities (exceeding twice the value of the *pho4Δ* control with and without ScPho2, dotted lines), a percentage value was calculated by dividing the activity without ScPho2 by that with ScPho2, after subtracting the negative control (*pho4Δ*) in both cases. All results are representative of multiple (>2) experiments.

## Reduced dependence on Pho2 is correlated with an increase in the number of Pho4 activated genes in *S. cerevisiae*

Since dependence on Pho2 provides additional selectivity for Pho4 induced gene expression in *S. cerevisiae* (*Zhou et al., 2011*), we hypothesized that a reduction in Pho2-dependence would result in an increase in the number of Pho4 targets in the *S. cerevisiae* background. To test this prediction, we quantified the number of genes induced by different Pho4 orthologs expressed in an *S. cerevisiae* background lacking the negative regulator Pho80, which allowed us to focus on Pho4-dependent genes but ignore starvation-induced, Pho4-independent genes (*Zhou et al., 2011*). We

identified a total of 247 genes that were significantly induced by at least one of the eight Pho4 orthologs in the presence of ScPho2 (False discovery rate < 0.05, fold change > 2). Pho4 from *C. glabrata*, *C. bracarensis*, *N. delphensis* and *C. nivariensis* induced more genes than the Pho2-dependent Pho4 orthologs did (*Figure 3A,D*), and genes induced by these Pho4 orthologs are largely unaffected when ScPho2 is absent (*Figure 3B*). For example, 212 genes were induced by CgPho4 with ScPho2, compared to 40 genes induced by ScPho4 in the same background. Pho4 from *S. paradoxus*, a close relative of *S. cerevisiae*, induced a smaller number of genes than ScPho4, as did Pho4 from *L. kluyveri*, an outgroup of both *S. cerevisiae* and *C. glabrata*. Thus, the observed target expansion is not congruent with the phylogenetic relationship, but is a property unique to Pho4 orthologs with reduced Pho2-dependence. Moreover, differences in the mean expression levels of the Pho4 orthologs are small (<2.5 fold, *Figure 3—figure supplement 1*) and do not explain the variation in their activity or dependence on Pho2 (*Figure 3—figure supplement 2*).

We further investigated whether there is a quantitative relationship between the level of Pho2-dependence and the number of genes induced by Pho4 orthologs. We previously measured the level of Pho2-dependence by comparing the activity of a single Pho4 target – *PHO5* – in a pair of strains differing in the presence or absence of ScPho2. Here we made the same comparison for gene induction fold changes in a group of 16 genes induced by all eight Pho4 orthologs (*Figure 3A*, red box). The results, measured by the mean of the ratios for the 16 genes, are largely consistent with what we observed with Pho5 alone (*Figure 3C*), and, in general, the number of genes induced by each Pho4 ortholog increases with decreasing levels of Pho2-dependence (*Figure 3D*). In summary, the level of Pho2-dependence is negatively correlated with the number of genes induced by the Pho4 ortholog in the *S. cerevisiae* background.

## CgPho4 binds to more locations in *S. cerevisiae* and activates a higher percentage of genes upon binding

We reasoned that the expansion of target genes for the *glabrata* clade Pho4 orthologs could result from Pho4 binding to more genomic locations, Pho4 activating a higher proportion of the genes to which it binds, or a combination of the two. To test if differences in Pho4 binding account for target gene expansion, we performed chromatin immunoprecipitation followed by high-throughput sequencing (ChIP-seq) to identify the binding locations for both ScPho4 and CgPho4 in the *S. cerevisiae* background lacking the negative regulator Pho80. We identified a total of 115 ChIP-peaks for CgPho4 and 74 peaks for ScPho4, with 72 being bound by both (*Figure 4A*, *Figure 4—source data 1*). The expansion of CgPho4 binding locations was not because it recognized new sequence motifs – 42 of the 43 CgPho4-specific peaks contain the consensus 'CACGTG' motif, and 69 of the 72 shared peaks contain this motif. For all four exceptions, a one-base-pair (bp) mismatch to the consensus motif is observed (*Figure 4A*, parentheses). Therefore, DNA binding specificity is conserved between ScPho4 and CgPho4. In contrast, ScPho4 and CgPho4 differ in their dependence of DNA binding on Pho2 – ScPho4 binding is significantly lower when ScPho2 is absent, but CgPho4 binding is largely unaffected by the deletion of ScPho2 (*Figure 4B*). In summary, CgPho4 recognizes the same E-box motif as ScPho4 does, but its binding is no longer dependent on ScPho2 and CgPho4 binds to ~50% more ($43/74 \approx 0.55$) sites than ScPho4 does.

It has been reported that both nucleosomes and another transcription factor with similar binding specificity to Pho4, Cbf1, competitively exclude ScPho4 from the E-box motifs in the genome (*Zhou et al., 2011*). It is therefore plausible that CgPho4 is able to bind to more locations because it can access sites normally occupied by nucleosomes or competitors. To test if nucleosome exclusion plays a role, we mapped the published nucleosome occupancy in phosphate-replete (high Pi) conditions (*Zhou et al., 2011*), where Pho4 is inactive, to the binding peaks identified in this study. We find that sites bound only by CgPho4 have on average higher nucleosome occupancy in high Pi conditions than sites bound by both CgPho4 and ScPho4 (*Figure 4C*, t-test for difference in the mean: *p*-value = 0.012, two-sided test). We performed the same analysis for Cbf1 enrichment in high Pi conditions at the top 25% most accessible E-box motifs – those with the lowest nucleosome occupancy – to avoid confounding nucleosome competition with Cbf1 binding. We found that Cbf1 enrichment at sites bound only by CgPho4 is not significantly different from enrichment at sites bound by both ScPho4 and CgPho4 (*Figure 4—figure supplement 1*). It is worth noting, however, that the small sample size in the CgPho4 bound only class (11) may have precluded us from detecting small differences. In conclusion, CgPho4 binding is less dependent on ScPho2 and it competes

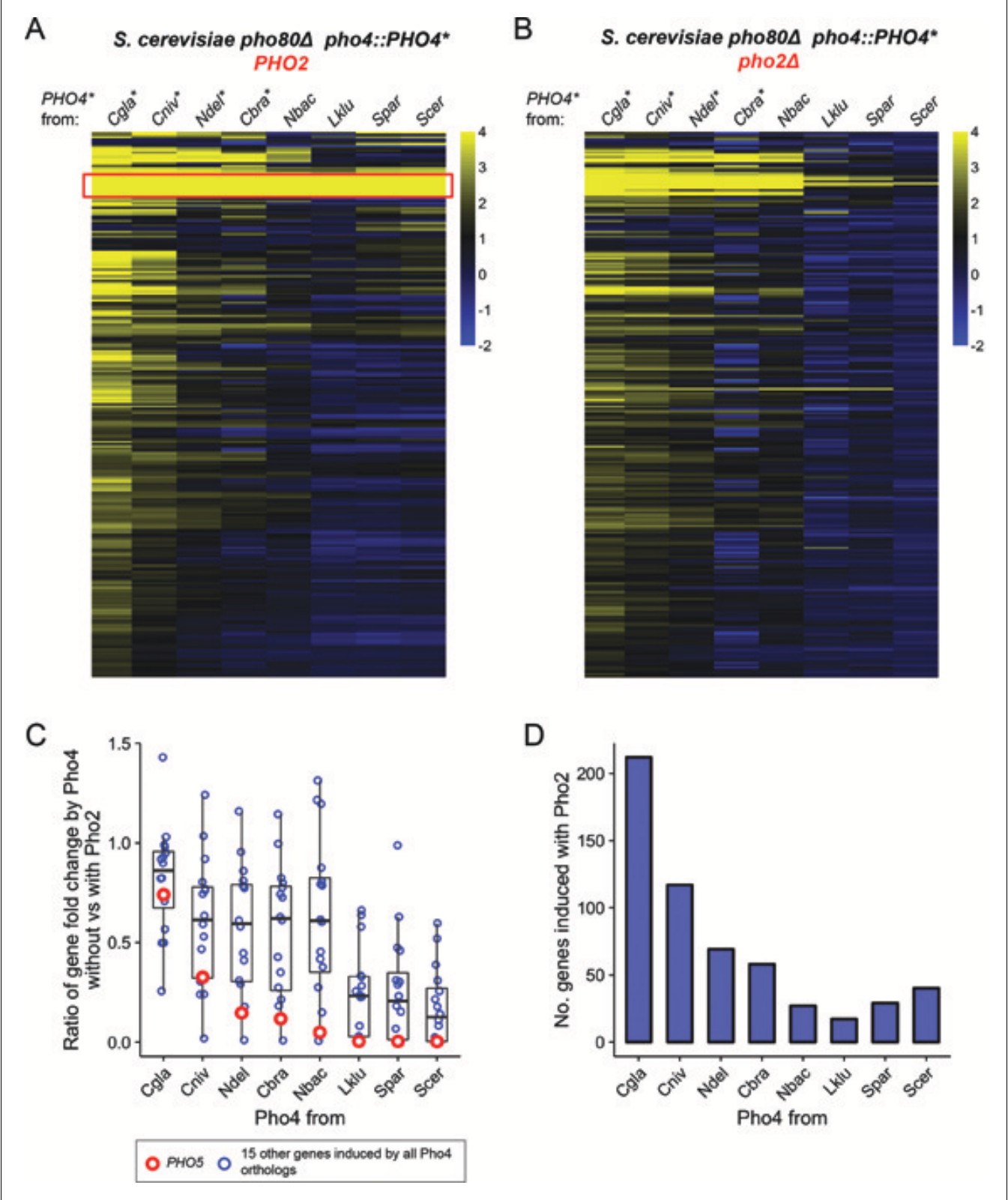

**Figure 3.** Pho4 orthologs that are less dependent on Pho2 induce more genes in the *S. cerevisiae* background. (**A**) Heatmap showing log$_2$ fold change of genes (rows) induced by Pho4 orthologs (columns) in the *S. cerevisiae* background lacking *PHO80*, with ScPho2. A cutoff of 4 and −2 are used for visual presentation. The raw fold change estimates for the 247 genes by eight Pho4 orthologs were available in *Figure 3—source data 1*. Species names for each of the Pho4 orthologs were abbreviated and correspond to the full names in *Figure 2*. An asterisk indicates the Pho4 ortholog was

*Figure 3 continued on next page*

*Figure 3 continued*

shown to induce Pho5 expression in the absence of ScPho2 in *S. cerevisiae*. A total of 247 genes are plotted. The red box highlights a group of 16 genes that were induced by all eight Pho4 orthologs tested. (B) Same as (A) except the strains were in a *pho80Δ pho2Δ* background for all Pho4 orthologs. (C) Scatter plot comparing the levels of Pho2-dependence for each Pho4 ortholog, measured by the ratios for the 16 shared target genes between their fold changes in the absence versus in the presence of ScPho2. The boxplots represent the interquartile range (IQC, box), the mean (thick bar in the middle) and the highest or lowest values within 1.5 times of IQC (whisker). The red circles highlight one of the 16 genes, *PHO5*. (D) Bar plot showing the number of genes significantly induced more than twofold by each Pho4 ortholog in the presence of ScPho2.

The online version of this article includes the following source data and figure supplement(s) for figure 3:

**Source data 1.** This zip file contains four tab-delimited csv files.
**Figure supplement 1.** Expression levels of the Pho4 orthologs by RNA-seq.
**Figure supplement 2.** Expression level of a Pho4 ortholog does not correlate with its level of Pho2-dependence or the number of genes it induces.

more favorably with nucleosomes than ScPho4 does, which likely contributes to the expansion of CgPho4 binding sites in the *S. cerevisiae* genome.

To compare the ability of CgPho4 and ScPho4 to induce gene expression upon binding to the promoter, we analyzed the transcriptional profiling data for the genes bound by the two Pho4 orthologs. We found that CgPho4 not only bound to more sites, but it also activated a higher percentage of the downstream genes upon its binding than did ScPho4 (64/115 = 56% vs 20/74 = 29%, *Figure 4D*). Moreover, its ability to induce gene expression is largely independent of Pho2: > 90% of CgPho4 targets (60/64 = 93.75%) were induced in the *pho2Δ* background, while all (20/20) ScPho4 induced genes required ScPho2 (*Figure 4D*).

In conclusion, CgPho4 both binds DNA and activates gene expression independently of ScPho2. Compared to ScPho4, it is more capable of accessing nucleosome-occluded binding motifs and it is also able to activate downstream gene expression with a higher probability upon binding. We propose that the combination of these features led to the expansion in the targets of CgPho4 in *S. cerevisiae* and speculate that this may underlie the target expansion for Pho4 from *C. bracarensis*, *N. delphensis* and *C. nivariensis*.

## In *C. glabrata* CgPho4 binds to DNA and activates gene expression largely independent of Pho2

Next we asked if CgPho4 also functions independently of Pho2 in its endogenous genome. To investigate the dependence of CgPho4 binding on CgPho2, we used the high resolution ChIP-exo technique to map CgPho4 binding in the presence and absence of CgPho2, and CgPho2 binding, under both phosphate-replete and phosphate-limited conditions (Materials and methods) (*Rhee and Pugh, 2012*). We identified a total of 100 binding peaks for CgPho4 in the presence of CgPho2 under phosphate-limited conditions (*Figure 5—source data 1*). CgPho4 recognizes the same 'CACGTG' motif as it does in *S. cerevisiae* (Materials and methods), and the consensus motif is present in 51 of the 100 peaks, with the rest containing a one-bp mismatch (46) or two-bp mismatches (3). CgPho2 bound to more than 500 sites genome-wide, without a strongly enriched sequence motif (Materials and methods). With respect to CgPho4 bound sites, CgPho2 binds at the same location for 77 of the 100 peaks (*Figure 5A*). Among these shared binding peaks, only 14 (18%) CgPho4 peaks showed more than two-fold reduction in peak height in the *pho2Δ* background (*Figure 5A*). We hypothesized that the quality of the DNA motif underlying the peak may explain the differential requirement of Pho2 co-binding. We tested this hypothesis by comparing changes in CgPho4 binding when CgPho2 is deleted, at sites with a consensus motif vs those without (*Figure 5—figure supplement 1*). Although the trend matches our expectation, the difference is small and not significant at a 0.05 level by the Student's t-test (*p*-value = 0.11). We conclude that CgPho4 binding to DNA is largely independent of CgPho2 in *C. glabrata*, but a small fraction (18%) of its binding sites show CgPho2 influence.

To evaluate whether gene induction by CgPho4 is dependent on CgPho2, we used RNA-seq to quantify the fold changes for genes induced by CgPho4 with or without CgPho2, in a strain lacking the negative regulator CgPho80. Intersecting with the ChIP identified CgPho4 binding sites, we identified 79 genes that were both directly bound and induced by CgPho4 in the presence of CgPho2 (*Figure 5—source data 2*). We then used mutant-cycle analysis (*Capaldi et al., 2008*) to delineate the contribution from either CgPho4 acting alone (Pho4), CgPho2 acting alone (Pho2) or

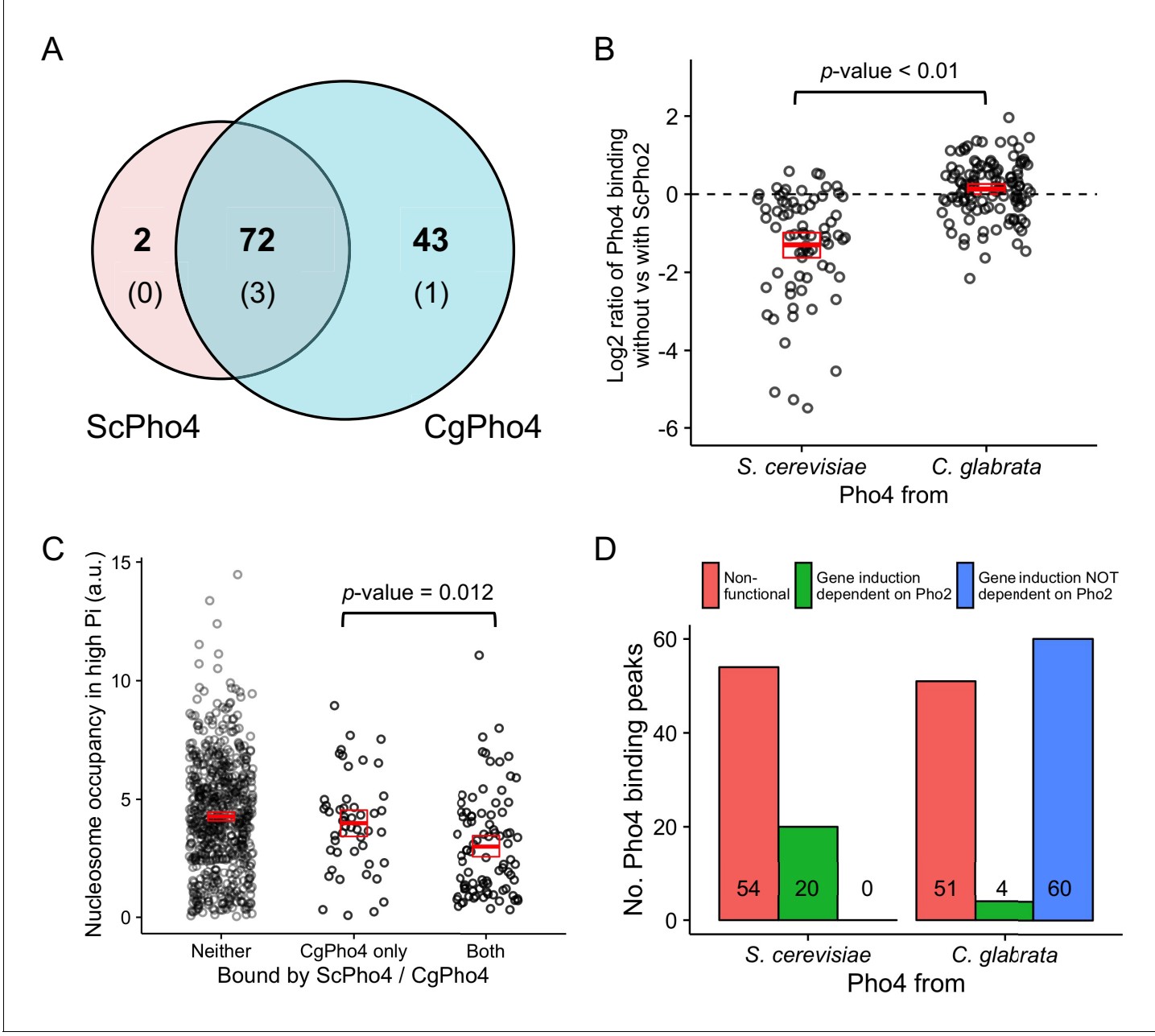

**Figure 4.** CgPho4 binds to more genomic locations than ScPho4 and is more likely to lead to gene activation upon binding in *S. cerevisiae*. (A) Venn diagram showing the number of and overlap between binding locations for ScPho4 and CgPho4 in the *S. cerevisiae* genome. The numbers in parentheses indicate binding events among the total number where the DNA sequence underlying the peak contains a suboptimal motif with one base pair mismatch to the consensus. (B) Scatter plot showing $\log_2$ ratio of ScPho4 or CgPho4 ChIP occupancy without vs with ScPho2. Only sites bound by Sc or CgPho4 in the presence of ScPho2 (N = 74 and 115, respectively) are plotted. The thick red bar represents the mean and the box the 95% confidence limits computed by a non-parametric bootstrapping method (*Harrell, 2016*). The means of the two groups are significantly different by a two-sided Student's t-test, with a *p*-value < 0.01. (C) Scatterplot for nucleosome occupancy in high Pi conditions at 'CACGTG' motifs either bound by neither ScPho4 nor CgPho4 (N = 660), only by CgPho4 (N = 48) or by both (N = 88). The red bar and box have the same meaning as in (B), and the difference between sites bound only by CgPho4 and those bound by both CgPho4 and ScPho4 is significant by a two-sided Student's t test (*p*-value = 0.012). (D) Bar plot comparing the number of genome-wide binding peaks for ScPho4 and CgPho4 that are either non-functional, lead to gene induction only with ScPho2 or lead to gene induction with or without ScPho2. The source data listing all identified ChIP peaks and the associated gene induction statistics are provided in *Figure 4—source data 1*.

The online version of this article includes the following source data and figure supplement(s) for figure 4:

**Source data 1.** List of ChIP-identified binding sites of ScPho4 and CgPho4 in *S. cerevisiae*, and associated gene information.

*Figure 4 continued on next page*

*Figure 4 continued*

**Figure supplement 1.** Cbf1 enrichment in high Pi conditions in *S. cerevisiae* is not significantly different between sites bound by both CgPho4 and ScPho4 and sites bound by CgPho4 only.

the two factors acting cooperatively (CO), and used unsupervised clustering (Ward's method, Materials and methods) on the estimated values for the three components to group the genes into three classes (*Figure 5B,C*). In class I (50 genes), CgPho4 is the dominant contributor to gene induction, with CgPho2 either contributing to a lesser extent by itself (Pho2) or through its interaction with CgPho4 (CO). Class II (11) genes show a strong collaborative component, with little contribution from Pho4 acting alone. Class III (18) genes show relatively low fold changes, with the main contribution coming from either CgPho4 acting alone or its collaborative effect with CgPho2. In conclusion, we found that >60% (50/79) of the genes bound by CgPho4 are induced primarily by CgPho4 acting alone, and that a lesser fraction (11/79) depend on the collaborative action of CgPho4 and CgPho2. This is in contrast to *S. cerevisiae*, where the majority of gene induction was attributed to the cooperative interaction between ScPho4 and ScPho2 (23/28 genes are induced only when both ScPho4 and ScPho2 are present [*Zhou et al., 2011*]).

## Pho4 direct targets in *C. glabrata* may function beyond phosphate homeostasis

In *S. cerevisiae*, nearly all ScPho4 targets function in either regulating the PHO pathway or maintaining intracellular phosphate homeostasis (*Ogawa et al., 2000*; *Zhou et al., 2011*). To gain insight into the function of the PHO pathway in *C. glabrata*, we studied the Gene Ontology (GO) terms associated with the 79 genes bound and induced by CgPho4 in *C. glabrata* (*Figure 5—source data 2*). The top three enriched GO terms for CgPho4 targets are related to phosphate homeostasis, i.e. polyphosphate metabolism, phosphorus metabolism and phosphate ion transport (*Figure 6—source data 1*), confirming that the PHO pathway in *C. glabrata* is conserved in its core function. However, two observations stand out. First, the genes in this core functional group are not all conserved (*Figure 6A*). Underlying the apparent conservation in function are non-orthologous genes that are either paralogs (e.g. *HOR2* in *S. cerevisiae* vs *RHR2* in *C. glabrata*) or evolutionarily unrelated (e.g. the phosphatase function of *PHO5* in *S. cerevisiae* is replaced by that of *PMU2* in *C. glabrata*) (*Figure 6A*, *Figure 6—source data 2*, [*Kerwin and Wykoff, 2009*, *2012*; *Orlando et al., 2015*]).

Second, genes with functions related to phosphate homeostasis account for only 16 of 79 CgPho4 targets. To predict the functions of the remaining CgPho4 targets, we mapped all 79 genes to GO Slim terms and identified several major functional groups (*Figure 6B*, *Figure 6—source data 3*). Pho4 targets in *C. glabrata* are enriched in genes predicted to be involved in the response to non-phosphate-related stresses, response to chemical stresses, fungal cell wall biosynthesis and cell adhesion, and carbohydrate metabolism (*Figure 6—source data 4*- Table S1-5). Thus, it appears that the PHO regulon in *C. glabrata* has, by expanding the number of targets, likely expanded its function beyond phosphate homeostasis. It is worth mentioning that, although CgPho4 induces more genes in *S. cerevisiae* as well as in *C. glabrata*, there is virtually no overlap between its targets in *S. cerevisiae* and in *C. glabrata*, except for those involved in phosphate homeostasis (as shown in *Figure 6A*). This is interesting because one might assume that the additional targets of CgPho4 in *C. glabrata* were evolutionarily acquired by exploiting existing E-box motifs in the genome that were inaccessible to the ancestral Pho2-dependent Pho4. This may still be true for some of the targets, as it is possible that the *S. cerevisiae* orthologs of those *C. glabrata* genes lost the E-box motifs after the two species diverged. The alternative hypothesis is that the targets not involved in phosphate homeostasis were gained de novo in *C. glabrata* by acquiring CgPho4-recognized motifs in their promoters. In conclusion, our comparative study of the Pho network between *S. cerevisiae* and *C. glabrata* reveals a substantial target expansion in the latter species. Comparative studies of more closely related species are needed to provide the temporal resolution for reconstructing the tempo and mode of target expansion in *C. glabrata*.

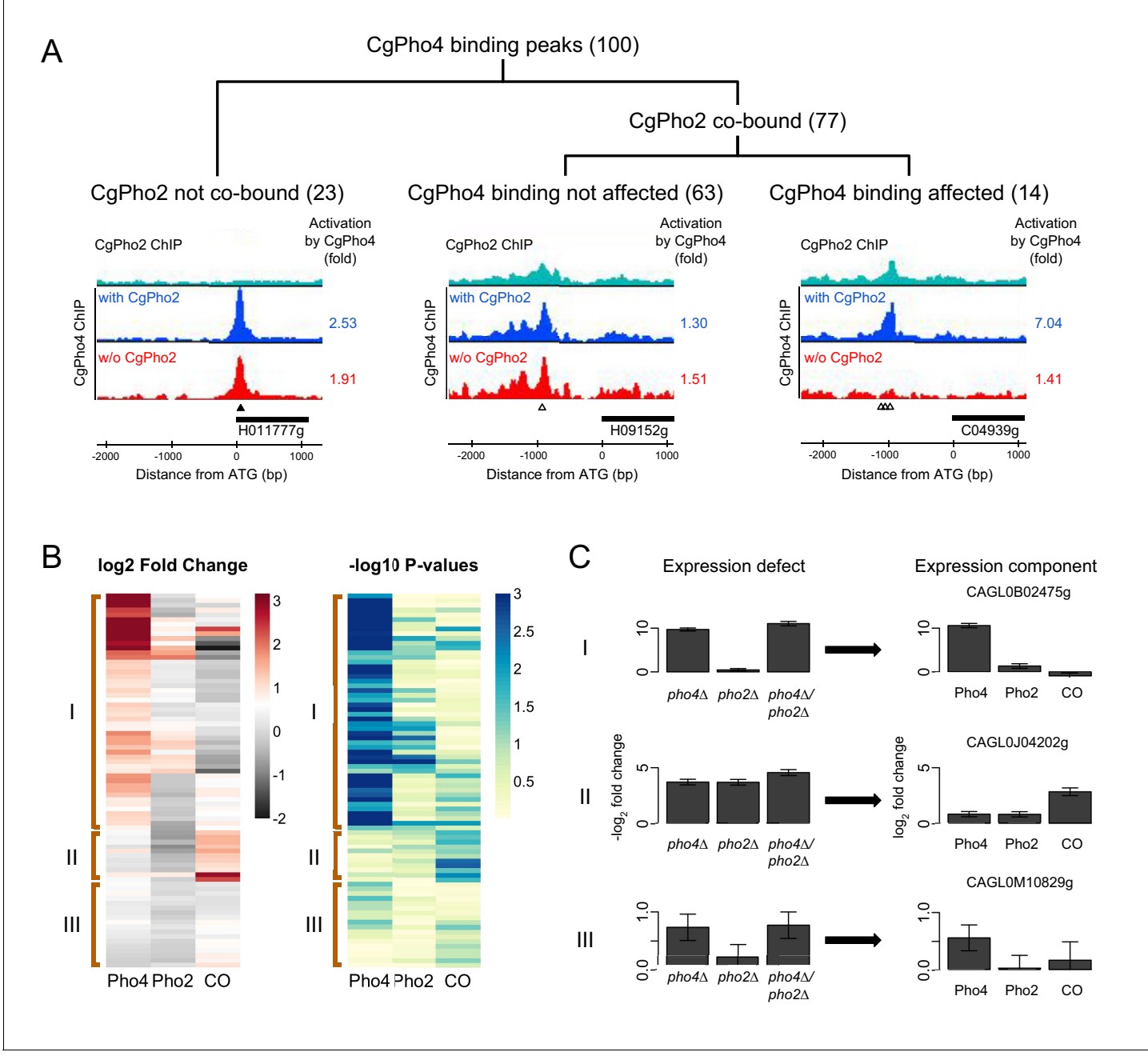

**Figure 5.** Identifying Pho4 targets in *C. glabrata* using genome-wide binding and transcriptome profiling data. (**A**) Dendrogram showing the breakdown of the 100 CgPho4 ChIP peaks based on whether CgPho2 binds next to CgPho4, and when it does, whether CgPho4 binding is affected by the deletion of Pho2 or not (defined as CgPho4 ChIP peak height reduced by more than twofold in the *pho2Δ* background). The graphs below the dendrogram show examples of ChIP profiles for each category of CgPho4 binding. Profiles of ChIP fold enrichment over mock are shown for CgPho2 in cyan, CgPho4 with CgPho2 in blue and CgPho4 without CgPho2 in red. The filled triangles indicate the location of a consensus "CACGTG" motif while the open triangles the one-bp-mismatches. The downstream gene is depicted as a thick bar to the right and the shortened systematic name (remove the preceding 'CAGL0') is shown below. The fold changes in induction for the putative target gene with or without CgPho2 are shown to the right of each graph. A list of the 100 binding peaks and the associated statistics are available in *Figure 5—source data 1*. (**B**) The left heat map showing the estimates of expression components for 79 genes directly bound and induced by CgPho4. For each gene, the $\log_2$ transformed fold change is decomposed into Pho4 effect alone (Pho4) + Pho2 effect alone (Pho2) + Pho4/Pho2 collaborative effect (CO). A cutoff of 3 and -2 is used for visual presentation. The unadjusted estimates and the associated *p*-values are available in *Figure 5—source data 2*. The right heatmap shows the $-\log_{10}$ transformed *p*-values for the corresponding t-statistics of the estimates on the left. Three groups are defined based on their characteristic expression components: group I genes are dominated by CgPho4 main effect; group II genes depend on both CgPho4 and CgPho2 (CO component); group III genes are a mix of the first two groups, with lower fold changes (weakly induced). (**C**) Bar graphs on the left showing the linear model estimates and the

*Figure 5 continued on next page*

*Figure 5 continued*

standard deviation of the expression defects, defined as the -log2 transformed fold changes between the mutants (single or double) and the wild-type. On the right are the corresponding estimates and standard deviation of the expression components for the same gene, estimated from the same data with two biological replicates per strain (Materials and methods). One representative gene is plotted for each category in (**B**).

The online version of this article includes the following source data and figure supplement(s) for figure 5:

**Source data 1.** List of ChIP-identified binding sites of CgPho4 in *C.glabrata*.
**Source data 2.** List of CgPho4 directly bound and induced genes and the associated expression components from the mutant cycle analysis.
**Figure supplement 1.** Comparison of CgPho2 influence on CgPho4 binding at sites with and without the consensus motif.

## Discussion

### Evolutionary constraint and plasticity in the PHO response network

GRNs consist of regulators, targets and the connections among them that may differ in their evolutionary plasticity. Our functional genomic comparison between the *S. cerevisiae* and *C. glabrata* PHO network reveals both constraints and plasticity in its evolution. In terms of constraint, the core transcriptional regulator Pho4 is conserved as a single copy gene and regulates the phosphate starvation response in distantly related species such as *S. cerevisiae* and *C. albicans* (*Ikeh et al., 2016*). Second, the mechanism for regulating Pho4 activity in response to phosphate starvation is conserved between *C. glabrata* and *S. cerevisiae* – in both species, Pho4 nuclear localization is phosphorylation-dependent, controlled by homologous cyclin-dependent kinase complexes (*Kerwin and Wykoff, 2012*). Third, both CgPho4 and ScPho4 recognize the E-box motif 'CACGTG' (Materials and methods). Together, these results show that the master TF – its identity, DNA-binding specificity and the mechanism of its regulation – are highly constrained during evolution. This is consistent with previous findings showing that the identity and sequence specificity of a TF evolve much more slowly compared to its target genes (*Wilson et al., 2008*).

What has changed throughout evolution in the PHO response network is the combinatorial regulation by Pho4 and Pho2, leading to changes in the network targets. Specifically, while *PHO2* as a gene is conserved among all 16 species examined, its functional role in the regulation of the PHO network has been dramatically reduced in *C. glabrata*. This, in turn, led to a dramatic expansion of the targets of CgPho4, potentially extending the function of the network beyond phosphate homeostasis. While the mechanism for this reduction in co-activator dependence is not clear, both transcriptional and binding assays are consistent with the hypothesis that CgPho4 evolved to be stronger in DNA-binding and in inducing gene expression (*Figure 4*).

Compared to previous findings in GRN evolution, two differences are worth noting. Unlike previous studies which found conserved network output despite regulatory rewiring (*Tsong et al., 2006*; *Kuo et al., 2010*; *Habib et al., 2012*), we observed a significant change in the size of the regulon (~20 target genes in *S. cerevisiae* vs.~70 in *C. glabrata*), despite a significantly shorter evolutionary time between the two focal species in this study compared to one of the previous studies (*Tsong et al., 2006*). This may be attributable to the different properties of the GRNs: previous studies have focused on developmental GRNs, whose target genes often constitute a cohesive module that function collectively to specify cell fates. By contrast, the target genes of stress response networks are more independently organized, either acting alone or in small subgroups, e.g. polyphosphate synthesis and phosphate transporters. The organization of a stress response GRN may allow it to be more evolutionarily plastic in acquiring and shedding targets as the environment shifts, creating new demands while eliminating old ones. Consistent with this view, previous studies showed that expression divergence evolves more readily among stress responsive genes compared to growth control and general metabolism genes (*Thompson and Regev, 2009*).

A second difference worth noting is the mechanism by which the target expansion occurred. Because the DNA sequence specificity of a TF is highly constrained during evolution (*Maerkl and Quake, 2009*; *Struhl, 1987*; *Wilson et al., 2008*; *Nitta et al., 2015*), it is traditionally thought that gain and loss of targets occurs primarily through *cis*-regulatory evolution via gain or loss of DNA motifs (*Peter and Davidson, 2011*; *Ihmels et al., 2005*; *Wittkopp and Kalay, 2011*). However, we find that *trans* evolution in CgPho4, which conserved the core DNA binding specificity and thus the existing targets, altered its dependence on the co-activator, and resulted in a dramatic expansion in

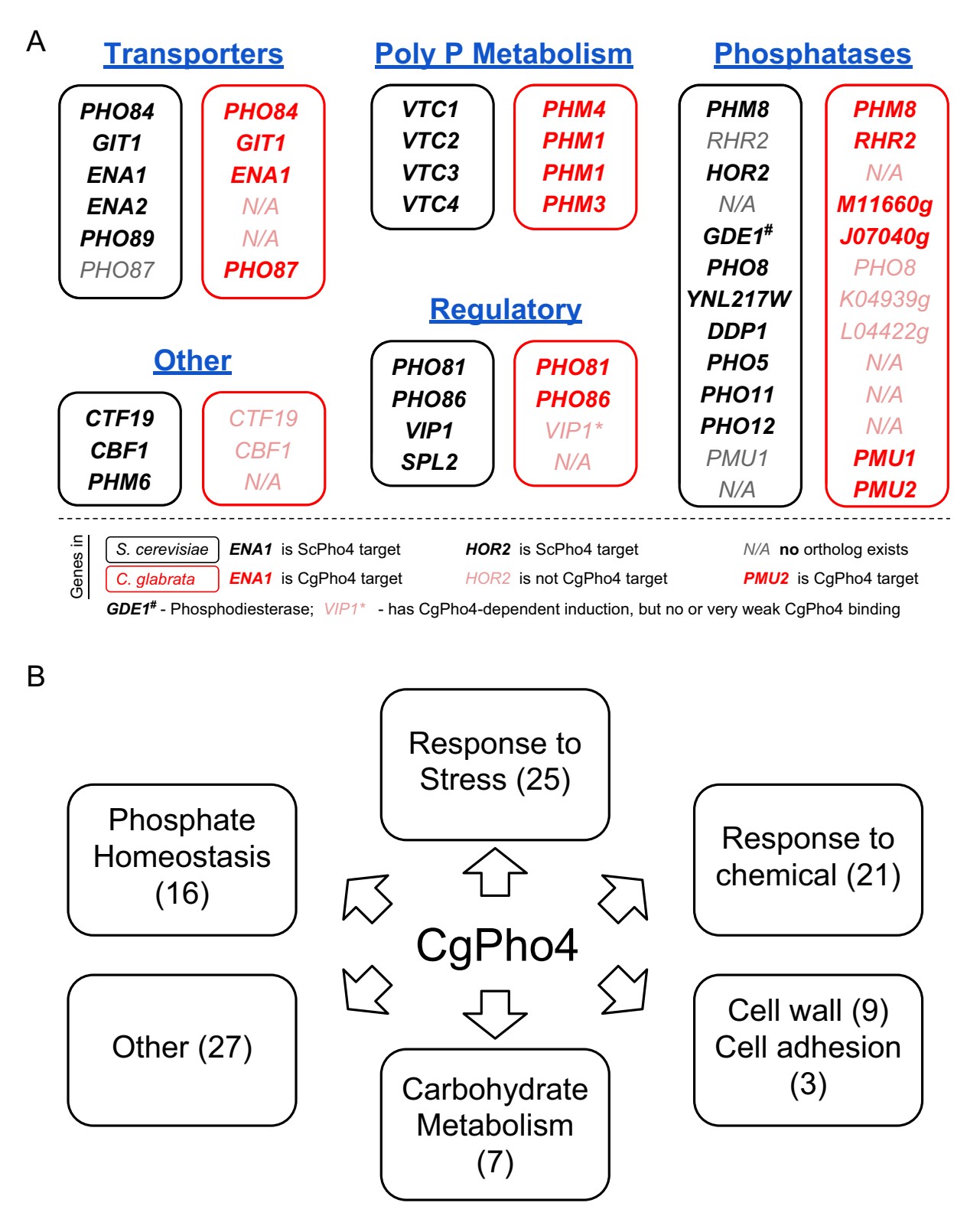

**Figure 6.** Functional annotation of Pho4 targets in *C. glabrata*. (**A**) Comparison between all 24 ScPho4 targets in *S. cerevisiae* and CgPho4 targets with phosphate homeostasis related functions in *C. glabrata*. Abbreviations: 'Poly P' stands for 'Polyphosphate'. Within each subcategory, *S. cerevisiae* genes in black are paired with their homologs in *C. glabrata* in red. N/A in either species indicates ortholog does not exist in that species. *C. glabrata* genes not annotated with a common name are represented by their systematic name with the preceding 'CAGL0' omitted. Gene names in bold
*Figure 6 continued on next page*

*Figure 6 continued*

indicate that they are targets of ScPho4 or CgPho4, while gray or pink gene names indicate they are not targets of ScPho4 or CgPho4, respectively. (B) Non-exclusive groups of Pho4 targets in *C. glabrata* based on Gene Ontology 'Biological process' terms and functional annotations in Candida Genome Database (for *C. glabrata* or orthologs in *C. albicans*) and Saccharomyces Genome Database (for orthologs in *S. cerevisiae*).

The online version of this article includes the following source data for figure 6:

**Source data 1.** Gene Ontology terms enrichment analysis results.
**Source data 2.** Table comparing ScPho4 and CgPho4 targets with phosphate homeostasis related functions.
**Source data 3.** All 79 CgPho4 targets in *C. glabrata* mapped to Gene Ontology Slim terms.
**Source data 4.** Contains five tables listing CgPho4 target genes annotation grouped by functional categories: Table S1 – Non-phosphate related stress and starvation response; Table S2 – Response to chemicals; Table S3 – Cell wall and cell adhesion; Table S4 – Carbohydrate metabolism; Table S5 – all other functional groups.

its targets. Compared to target turnover via *cis* regulatory changes, this *trans* evolutionary mechanism is likely much more rapid, and may allow natural selection to quickly sample many potential targets at once. However, it should be noted that the reduction in Pho2-dependence likely evolved gradually (*Figures 2B* and *3C*), which means *cis* evolution could have accompanied the *trans* – while Pho4 expands its targets, promoter evolution could result in fixation of beneficial targets while removing spurious, non-beneficial ones. Thus, the full picture is likely far more complex.

In conclusion, we demonstrated that evolution of combinatorial regulation can lead to rapid rewiring of a gene regulatory network. In the PHO network, this could provide a convenient 'switch' for evolution to rapidly alter the size of the network output instead of relying on individual promoter alterations. More generally, changes affecting the interaction between transcription factors and their cofactors may be an important yet underappreciated mechanism underlying gene regulatory network evolution (*Slattery et al., 2011*).

## Expansion in the function of the PHO regulon and its implication for the stresses faced by *C. glabrata*

Only 16 of the 79 genes directly induced by CgPho4 in *C. glabrata* are involved in maintaining phosphate homeostasis. Among the remaining genes, a significant number (25 genes) are predicted to be involved in responses to non-phosphate related stresses, including osmotic (seven genes) and oxidative (seven genes) stresses. In addition to roles in other stress responses, a smaller group of CgPho4 targets (9 and 3 genes) have potential functions in cell wall synthesis and cell adhesion, two traits that were known to be relevant for survival and virulence in the host (*Atanasova et al., 2013*; *De Las Peñas et al., 2015*; *Luo and Samaranayake, 2002*; *Jawhara et al., 2012*; *Fabre et al., 2014*). A similar observation has been made before in *C. glabrata*, where limitation of nicotinic acid was sufficient to induce genes mediating a cellular adhesion phenotype (*Domergue et al., 2005*). Intriguingly, Pho4 in *C. albicans* has been shown to be important for survival under particular types of osmotic and oxidative stresses, suggesting a similar functional expansion as we observed in *C. glabrata* (*Ikeh et al., 2016*). Also in *C. albicans*, phosphate starvation was linked to enhanced virulence, and a strain lacking Pho4 displayed extensive filamentation in response to phosphate limitation (*Romanowski et al., 2012*). Combining these observations, we speculate that stress responses in the commensal species may have evolved to be more coordinated, and may have acquired new targets linked to virulence, to cope with the distinct stress profiles in the host, such as spatiotemporally overlapping challenges exerted by the host immune cells (*Kasper et al., 2015*). Characterizing other stress response pathways in *C. glabrata*, and in other commensal species such as *C. albicans*, can test this hypothesis and will shed further light on the architectural differences in the stress and starvation response network compared to that in the free-living species.

## Materials and methods

### Strains

#### S. cerevisiae strains

All *S. cerevisiae* strains were generated from EY0057 (K699 *ade2-1 trp1-1 can1-100 leu2-3,112 his3-11,15 ura3* GAL+), using a standard high efficiency transformation protocol with ~ 40 bp homology sequences (*Gietz and Schiestl, 2007*). The Pho4 ortholog swap strains (EY2863 etc., *Table 1*) were made by the *URA3* pop-in and pop-out method. Briefly, a *K. lactis URA3* gene with proper promoter and terminator was used to precisely replace the CDS of *PHO4* in EY2849, where *PHO80* is deleted. The *URA3* gene was then replaced by the CDS sequence from one of the *PHO4* orthologs, obtained via PCR-based cloning from the genomic DNA of the relevant species. Successful transformants were validated using PCR. *PHO2* was knocked out using *KlacURA3*, generating the complementary set of *pho2Δ* strains. Genomic DNA for non-*cerevisiae* species was prepared using standard methods (ball milling to break the cells, followed by Phenol Chloroform extraction and RNase treatment). PCR primers were designed based on genome sequences for each species from the Orthogroup website (*Wapinski et al., 2007*), except for *C. nivariensis*, *C. bracarensis*, *N. bacillisporus*, *N. delphensis* and *C. castellii*, which were initially provided to us by Dr. Cecile Fairhead and are now available through GRYC (Genome Resources for Yeast Chromosomes, http://gryc.inra.fr). Strains of the non-*S. cerevisiae* species were from Aviv Regev lab (*Thompson et al., 2013*) and Cecile Fairhead (for the *Nakaseomyces* genus). Abbreviations for species names are as follows: *S. mikatae (Smik), S. paradoxus (Spar), C. glabrata (Cgla), C. nivariensis (Cniv), C. bracarensis (Cbra), N. bacillisporus (Nbac), N. delphensis (Ndel), C. castellii (Ccas), N. castellii (Ncas), L. waltii (Lwal), L. kluyveri (Lklu), K. lactis (Klac), D. hansenii (Dhan), C. albicans (Calb), Y. lipolytica (Ylip)*. Species names were based on YGOB and GRYC (*Byrne and Wolfe, 2005*; *Gabaldón et al., 2013*). Bio-ChIP strains for either ScPho4 or CgPho4 were generated by replacing the *URA3* marker used to knockout the *PHO4* CDS with a linear DNA construct containing either ScPho4 or the CDS of CgPho4 fused with the Avitag (GLNDIFEAQKIEWHW) at the C-terminus, separated by a 'GSGSGS' linker. Primer sequences for generating the strains are available upon request.

#### C. glabrata strains

CG1 was generated from BG99 (*Cormack and Falkow, 1999*) by inactivating the *URA3* gene with a random piece of DNA amplified from pUC19 plasmid and selecting on 5-FOA medium. CG1 was subsequently used to generate the rest of *C. glabrata* strains using homologous recombination with either antibiotic resistance or nutrient markers. The *NAT* gene conferring resistance to clonNat was amplified from the pFA6a-natNT2 plasmid in the PCR Toolbox collection (*Janke et al., 2004*). A standard *S. cerevisiae* transformation protocol was used with the following modification: instead of the short 40–60 bp flanking sequences, long flanking homologous sequences between 100–1000 bp were used on each end. The transformation constructs were generated using 2-step PCR with the split *URA3* marker (*Reid et al., 2002*). ChIP-exo strains were made the same way as in *S. cerevisiae*, except using a C-terminal 3xFLAG tag (DYKDHDGDYKDHDIDYKDDDDK) for PHO4 with 6xG linker, and a N-terminal V5 tag (GKPIPNPLLGLDST) for PHO2 with 5xGS linker.

#### C. albicans strains

All *C. albicans* strains used in this study are from the transcriptional regulatory deletion library made by Homann et al (*Homann et al., 2009*) and ordered from the Fungal Genetics Stock Center (http://www.fgsc.net/, RRID:SCR_008143)

### Media and growth conditions

Phosphate-free synthetic complete medium was prepared from Yeast Nitrogen Base with ammonium sulfate, without phosphates, without sodium chloride (MP Biomedicals, Santa Ana, California) and supplemented to a final concentration of 2% glucose, 1.5 mg/ml potassium chloride, 0.1 mg/ml sodium chloride and amino acids, as described previously (*Lam et al., 2008*). Monobasic potassium phosphate (1M solution, Sigma-Aldrich, St. Louis, MO) was added to phosphate-free medium to make high phosphate (Pi) medium containing a final concentration of 10 mM Pi. All media were adjusted to pH 4.0 with HCl. Yeast strains were grown at 30°C with shaking and cell samples were

**Table 1.** Strains used in this study.

| Strain | Genotype | Reference |
|---|---|---|
| *S. cerevisiae* | | |
| EY0057 | K699, as wild-type for PHO pathway | (*O'Neill et al., 1996*) |
| EY0252 | *pho2::LEU2* | (*Kaffman et al., 1994*) |
| EY1710 | *pho4::URA3* | (*Zhou et al., 2011*) |
| EY2849 | *pho80::TRP1* | This study |
| EY2851 | *pho80::TRP1 pho2::HIS5* | This study |
| EY2852 | *pho80::TRP1 pho4::URA3* | This study |
| EY2859 | *pho80::TRP1 pho4::URA3 pho2::HIS5* | This study |
| EY2863 | *pho80::TRP1 pho4::CglaPHO4* | This study |
| EY2872 | *pho80::TRP1 pho4::SparPHO4* | This study |
| EY2873 | *pho80::TRP1 pho4::SmikPHO4* | This study |
| EY2874 | *pho80::TRP1 pho4::LkluPHO4* | This study |
| EY2875 | *pho80::TRP1 pho4::NcasPHO4* | This study |
| EY2876 | *pho80::TRP1 pho4::KlacPHO4* | This study |
| EY2877 | *pho80::TRP1 pho4::LwalPHO4* | This study |
| EY2878 | *pho80::TRP1 pho4::DhanPHO4* | This study |
| HY107 | *pho80::TRP1 pho4::CnivPHO4* | This study |
| HY108 | *pho80::TRP1 pho4::CbraPHO4* | This study |
| HY110 | *pho80::TRP1 pho4::NbacPHO4* | This study |
| HY111 | *pho80::TRP1 pho4::NdelPHO4* | This study |
| HY120 | *pho80::TRP1 pho4::CcasPHO4* | This study |
| HY132 | *pho80::TRP1 pho4::CalbPHO4* | This study |
| HY136 | *pho80::TRP1 pho4::YlipPHO4* | This study |
| EY2879 | *pho80::TRP1 pho4::CglaPHO4 pho2::URA3* | This study |
| EY2880 | *pho80::TRP1 pho4::SparPHO4 pho2::URA3* | This study |
| EY2881 | *pho80::TRP1 pho4::SmikPHO4 pho2::URA3* | This study |
| EY2882 | *pho80::TRP1 pho4::LkluPHO4 pho2::URA3* | This study |
| EY2883 | *pho80::TRP1 pho4::NcasPHO4 pho2::URA3* | This study |
| EY2884 | *pho80::TRP1 pho4::KlacPHO4 pho2::URA3* | This study |
| EY2885 | *pho80::TRP1 pho4::LwalPHO4 pho2::URA3* | This study |
| EY2886 | *pho80::TRP1 pho4::DhanPHO4 pho2::URA3* | This study |
| HY112 | *pho80::TRP1 pho4::CnivPHO4 pho2::HIS5* | This study |
| HY114 | *pho80::TRP1 pho4::CbraPHO4 pho2::HIS5* | This study |
| HY115 | *pho80::TRP1 pho4::NbacPHO4 pho2::HIS5* | This study |
| HY116 | *pho80::TRP1 pho4::NdelPHO4 pho2::HIS5* | This study |
| HY121 | *pho80::TRP1 pho4::CcasPHO4 pho2::HIS5* | This study |
| HY134 | *pho80::TRP1 pho4::CalbPHO4 pho2::HIS5* | This study |
| HY138 | *pho80::TRP1 pho4::YlipPHO4 pho2::HIS5* | This study |
| EY2681 | *pho80::HIS3 pho4::ScerPHO4-C-AVI-TRP1 ura3::pRS306-BirA* | (*Zhou et al., 2011*) |
| EY2869 | *pho80::TRP1 pho4::CglaPHO4-C-AVI ura3::pRS306-BirA* | This study |
| EY2867 | *pho80::TRP1 pho4::CglaPHO4-C-AVI pho2::URA3 ura3::pRS306-BirA* | This study |
| HY130 | *pho80::TRP1 pho4::CglaPHO4 ura3::pRS306-BirA* | This study |
| *C. glabrata* | | |

*Table 1 continued on next page*

Table 1 continued

| Strain | Genotype | Reference |
|---|---|---|
| BG99 | his3Δ (1 + 631), as wild-type for PHO pathway | (Cormack and Falkow, 1999) |
| HY1 | BG99, ura3Δ | This study |
| HY3 | pho80::HIS3 | This study |
| HY5 | pho4::URA3 | This study |
| HY6 | pho80::HIS3 pho4::URA3 | This study |
| HY8 | pho2::URA3 | This study |
| HY10 | pho80::HIS3 pho2::URA3 | This study |
| HY29 | pho80::HIS3 pho4::natNT2 pho2::URA3 | This study |
| HY68 | pho2::N-3xFLAG-PHO2 | This study |
| HY70 | pho2::N-V5-PHO2 | This study |
| HY75 | pho4::PHO4-C-3xFLAG | This study |
| HY85 | pho4::PHO4-C-3xFLAG pho2::URA3 | This study |
| HY89 | pho80::HIS3 pho2::N-V5-PHO2 | This study |
| C. albicans | | |
| Wild-type | LEU2 / leu2Δ HIS1 / his1Δ | (Homann et al., 2009) |
| orf19.1253 | pho4::LEU2 / pho4::HIS1 | (Homann et al., 2009) |
| orf19.4000 | pho2::LEU2 / pho2::HIS1 | (Homann et al., 2009) |

collected at early/mid-logarithmic phase ($OD_{600}$ 0.3–0.4). To induce the phosphate starvation response, yeast cells were first grown in 10 mM Pi medium to early/mid-logarithmic phase. Cells were then harvested by filtering and washed with 2–3 volumes of no Pi medium pre-warmed to 30°C. Finally, cells were re-suspended in pre-warmed no Pi medium and grown at 30°C for 1 hr before being harvested for downstream analyses. For plate growth assays, yeast cells were grown in 10 mM Pi medium until mid-logarithmic phase, washed 2–3 times and re-suspended in sterile water. A 1:4 dilution series were made with sterile water in 96-well plates. A 48-pin tool was used to transfer ~10 µL of resuspended cell culture onto appropriate solid agar plates. After 24–48 hr of growth at 30°C, pictures were taken using a standard gel box apparatus.

## Phosphatase assays

For the semi-quantitative assay, cells were grown overnight (preconditioning), diluted to $OD_{600}$ ~0.1 in the morning and grown to $OD_{600}$ = 0.6–1. The cell culture was centrifuged, washed and re-suspended in water. A four-fold serial dilution was prepared for each strain and spotted onto an agar plate using a 48-pin tool. Complete synthetic medium was used for pho80Δ strains and phosphate-free medium (see above) used for PHO80 wild-type strains. After overnight growth, the agar plates were overlaid with Fast Blue Salt B stain (Sigma-Aldrich D9805), 1-naphthyl phosphate (1 NP, Sigma-Aldrich, D5602), and 1% agar in 0.1 m sodium acetate (pH 4.2) (Wykoff et al., 2007). Pictures were taken on a HP color scanner after 5 min.

For the quantitative phosphatase assay, cells were preconditioned and grown the same way as above. After collection by centrifugation, cells were washed and re-suspended with sterile water to $OD_{600}$ ~ 5. Then 30 µL of the re-suspended culture was transferred to a 96-well assay plate in triplicates. The cell re-suspension was incubated with 80 µL of 10 mM p-nitrophenyl phosphate (pNPP, Sigma-Aldrich P4744) dissolved in 0.1M sodium acetate (pH = 4.2) for 15 min at 25°C. The reaction was quenched by adding 144 µL of saturated $Na_2CO_3$ (pH > 11) followed by 5 min of centrifugation at 3000 g. Finally, 200 µL of the supernatant from each well was transferred to a new plate and

$OD_{420}$ was measured on a plate reader. Phosphatase activity was measured in units expressed as $OD_{420}/OD_{600}$ (*Huang et al., 2005*).

## Transcriptional profiling by RNA-seq

All transcriptional profiling experiments were done in the *pho80Δ* background in both *S. cerevisiae* and *C. glabrata*. Two biological replicates (same genotype, but grown, collected and processed separately) were obtained for each sample. Briefly, yeast cells were collected using a cold methanol quenching method (*Pieterse et al., 2006*; *Zhou et al., 2011*). 20 mL of mid-log phase ($OD_{600}$ = 0.2–0.5) cell cultures were added directly into 30 mL of pre-chilled methanol ($\sim-50°C$), and incubated in an ethanol-dry ice bath at the same temperature for at least 20 min. Cells were collected by centrifugation and quickly washed with ice-cold water to remove methanol, and resuspended in RNAlater solution (Qiagen, Hilden, Germany). For each sample, $5*10^7$ cells were collected and mechanically lysed on a Minibeadbeater (BioSpec Products, Bartlesville, OK): Zirconia beads (0.5 mm, BioSpec Products #11079105z) were added to ~600 μL of cell suspension per sample in a 2 mL screw cap tube to the meniscus. Cells were lysed by four rounds of 1 min bead beating and 2 min of cooling in an ice-water bath. An RNasy Mini kit (Qiagen) was used to isolate total RNA from the lysed cell. RNA-seq libraries were prepared with the TruSeq RNA Library Preparation Kit v2 (Illumina, San Diego, CA) with the mRNA purification option, following the manufacturer's protocol. The resulting libraries were sequenced on an Illumina HiSeq 2000, which produced on average 10 million 50 bp single end reads for each sample.

## Chromatin immunoprecipitation

### Biotin-tag chromatin immunoprecipitation (Bio-ChIP) in *S. cerevisiae*

Bio-ChIP was modified from techniques previously described (*Zhou et al., 2011*; *Lam et al., 2008*; *Kolodziej et al., 2009*; *van Werven and Timmers, 2006*), and performed for ScPho4 and CgPho4 in *S. cerevisiae* with the negative regulator ScPho80 deleted (*pho80Δ*). In addition, CgPho4 ChIP was also performed in *S. cerevisiae* with both ScPho80 and ScPho2 deleted (*pho80Δ pho2Δ*). One biological sample was analyzed for each strain. Two types of controls were included, i.e. an input sample, which is sonicated chromatin not subject to immune-precipitation (IP), and a mock sample, which uses a strain lacking the epitope tag recognized by the antibody, and is subject to the same IP as done for the biological sample.~100 $OD_{600}$ units of early log phase cells ($OD_{600}$ ~ 0.3) were collected in phosphate-replete (10 mM Pi) conditions. Cells were cross-linked with 1% formaldehyde (Fisher-Scientific #AC41073-1000, Hampton, NH) for 10 min and then quenched with 125 mM glycine for 10 min at room temperature. Cells were collected by centrifugation for 15 min at 6000 rpm in an Avanti J-20 XP high speed centrifuge, with JLA10.500 rotor (Beckman Coutler, Brea, CA), immediately washed with cold PBS buffer (137 mM NaCl, 2.7 mM KCl, 10 mM Sodium phosphate dibasic, 2 mM potassium phosphate monobasic, pH 7.4), and mechanically lysed with 0.5 mm Zirconia beads in lysis buffer (50 mM HEPES, pH 7.5, 140 mM NaCl, 1 mM EDTA, 1% Triton X-100, 0.1% Na- Deoxycholate). Chromatin was fragmented by sonication using the Covaris E220 Adaptive Focus system (Covaris, Woburn, MA) using the 130 μL tube and the following setting: Duty Factor: 10%; Peak Incident Power: 175; Cycles/Burst: 200; Time:150s (*Elfving et al., 2014*). The cell lysate was then incubated with Dynabeads MyOne Streptavidin C1 (Invitrogen, Carlsbad, CA) overnight at 4°C. Afterwards, the dynabeads were washed with lysis buffer, high salt wash buffer (50 mM HEPES, pH 7.5, 500 mM NaCl, 1 mM EDTA, 1% Triton X-100, 0.1% Na-Deoxycholate), lithium wash buffer (10 mM Tris/HCl, pH 8.0, 500 mM LiCl, 1 mM EDTA, 1% NP-40, 0.1% Na-Deoxycholate), and SDS wash buffer (10 mM Tris/HCl, pH 8.0, 1 mM EDTA, 3% SDS) for 2 × 2 min at room temperature, and 1 × 2 min with TE buffer (10 mM Tris/HCl, pH 8.0, 1 mM EDTA). Crosslinking was reversed by incubation of samples at 65°C for at least 6 hr in 10 mM Tris-HCl, pH 8.0, 1 mM EDTA, 0.8% SDS. RNA and proteins in the samples were digested with 20 μg/ml RNase A (Thermo Fisher Scientific, Waltham, MA) at 37°C for 2 hr and 0.2 mg/ml proteinase K (Roche, Basel, Switzerland) for 2 hr at 55°C. DNA was then purified with minElute column (Qiagen). 5% of the volume of cell lysate was removed after sonication and used to prepare the input DNA for each ChIP experiment. ChIP libraries were prepared using the NEBNext Ultra II DNA Library Prep Kit (New England Laboratory, Ipswich, MA) and sequenced on an Illumina HiSeq2000 instrument to produce ~10 million 50 bp single end reads per sample.

## Chromatin immunoprecipitation exonuclease (ChIP-exo) in *C. glabrata*

ChIP-exo was performed in *C. glabrata* for C-3xFLAG tagged CgPho4 either with or without CgPho2, and N-V5 tagged CgPho2 following published methods (*Rhee and Pugh, 2012*; *Serandour et al., 2013*). All strains have wild-type CgPho80. Only mock samples, but not input samples, were included because the latter didn't apply to the ChIP-exo procedure (*Rhee and Pugh, 2012*). One biological sample was analyzed for each strain. Briefly, cells were grown to early log phase, at $OD_{600}$ ~ 0.3. For phosphate starvation samples, cells were filter collected, washed with equal volume of warm, phosphate-free media and released into fresh, phosphate-free media. Starvation continued for 1 hr and cells were fixed and processed in the same way as in Bio-ChIP until the sonication step. We also collected cells in phosphate-replete conditions. The same procedure as above was used, except cells were washed and released into high phosphate media instead of phosphate-free media. Afterwards, either the Anti-FLAG M2 Magnetic Beads (Sigma-Aldrich #M8823) or the Anti-V5 antibody (Abcam #15828, Cambridge, UK, RRID:AB_443253) was incubated with the cell lysates for 2 hr up to overnight. For Anti-V5 antibody, Dynabeads Protein G (Thermo Fisher Scientific #10003D) was added to the antibody-lysate mix and incubated for an additional 2 hr at 4℃. The following steps are modified from the Active Motif ChIP-exo vA4 manual (https://www.activemotif.com/documents/1938.pdf): with the chromatin still bound to the beads, the DNA was end-polished and P7-exo adapters are ligated onto the blunt ends. The nicked DNA was repaired and then digested by lambda and RecJf exonulceases to excise DNA in a 5' to 3' direction, trimming up to the site of the cross-linking and selectively eliminating the P7 adapter at the 5´end. Following cross-link reversal and elution from the beads, the DNA was made double-stranded by P7 primer extension and a P5-exo adapter was added to the exonuclease-treated ends. The DNA library was PCR amplified and size selected before it was subjected to high-throughput sequencing.

## Functional genomics data analysis

### RNA-seq: differential expression and mutant cycle analysis

#### Reads mapping and counting

The raw reads were mapped to the corresponding species reference genomes (*S. cerevisiae* genome version S288C_R64-1-1; *C. glabrata* genome version s02-m02-r09) using Bowtie (RRID:SCR_005476, v1.1.1) with the option '-m 1 –best –strata' (*Langmead et al., 2009*). The resulting SAM files were sorted using Samtools (RRID:SCR_002105, v1.2) (*Li et al., 2009*) and the number of reads per transcript was counted using Bedtools2 (RRID:SCR_006646) (*Quinlan and Hall, 2010*), with the option 'bedtools coverage -a BAM_file -b genome_annotation.bed -S -s sorted -g Chrom.length'. Gene features for *S. cerevisiae* were downloaded from the Saccharomyces Genome Database (SGD, RRID:SCR_004694), version R64-2-1; gene features for *C. glabrata* were downloaded from the Candida Genome Database (CGD, RRID:SCR_002036), version s02-m07-r04. The count matrix was imported into R for downstream analyses.

#### Differential gene expression analysis in *S. cerevisiae*

First, 676 genes having <1 count per million reads in at least three samples were removed, leaving 5707 genes in the dataset. Next, we used the trimmed mean of M-values ('TMM') method in the EdgeR (RRID:SCR_012802) package to calculate the normalization factors for scaling the raw library sizes (*Robinson and Oshlack, 2010*; *Robinson et al., 2010*), and applied voom transformation (*Law et al., 2014*) to remove the mean-variance relationship on the $\log_2$ transformed count data.

To identify genes induced by each Pho4 ortholog in the *S. cerevisiae* background, we compared the transcriptional profiles between a strain with the Pho4 ortholog and a strain without Pho4, either with or without *ScPHO2*, e.g. *CgPHO4 ScPHO2* vs *CgPHO4 pho2Δ*. Two biological replicates were analyzed for each strain. With eight Pho4 orthologs, there are a total of 16 pairwise comparisons for each gene in the dataset. In practice, we analyzed all 16 comparisons with a single linear model framework using the LIMMA (RRID:SCR_010943) package in R (*Ritchie et al., 2015*). The normalized $\log_2$ read count for gene $i$, strain $j$ and replicate $k$ can be expressed as $y_{i,j,k} = \beta_{i,j} + \varepsilon_{i,j,k}$, , where $j$ = {*pho4Δ.pho2Δ, pho4Δ.ScPHO2, PHO4*.pho2Δ, PHO4*.ScPHO2*}, with *PHO4** replaced by one of the eight Pho4 orthologs, and $k$ = 1, 2. The fold induction for gene $i$ by ScPho4 in the presence of ScPho2 can then be estimated as $\beta_{i,ScPHO4.ScPHO2} - \beta_{i,pho4Δ.ScPHO2}$. To test for significant differences in the expression of gene $i$ between two genotypes, LIMMA performs a moderated t-test, which differs

from a standard t-test in that it uses an Empirical Bayes method to moderate the standard error term for each gene so as to make the estimates more robust (*Smyth et al., 2005*). After obtaining the 16 fold induction estimates for each gene (8 Pho4 orthologs with or without ScPho2), we tested for significant induction against a null hypothesis of *fold change <= 2*, with the function treat(..., lfc = 1). The raw *p*-values were then *pooled* across all genes and all pairwise comparisons, and a Benjamini-Hochberg procedure was used to control the false discovery rate at 0.05, with the function decideTests(..., p.value=0.05, method = 'global') in LIMMA. The advantage of this approach is that the raw *p*-value cutoff is consistent across all 16 comparisons.

The estimated fold changes were used to produce *Figure 3A and B*, for the 247 genes that were significantly induced more than two-fold by at least one of the Pho4 orthologs. For *Figure 3C*, we calculated the ratios between the fold induction without vs with ScPho2 for each Pho4 ortholog, for the 16 genes significantly induced by all eight Pho4 orthologs. For *Figure 4C*, the number of genes significantly induced by each Pho4 ortholog with ScPho2 was determined by the method described above.

## Mutant cycle analysis in *C. glabrata*

In order to explicitly evaluate the contribution from CgPho4, CgPho2 and their collaboration to gene induction in *C. glabrata*, we applied the mutant cycle analysis method (*Capaldi et al., 2008*; *Zhou et al., 2011*), which decomposes the induction fold change of each gene into Pho4 acting alone (Pho4), Pho2 acting alone (Pho2) and Pho4/Pho2 collaborative effect (CO). To estimate the three components, we used the same linear model approach as in *S. cerevisiae*, where the fold induction for any gene $i$ in strain $j$ and replicate $k$ is expressed as $y_{i,j,k} = \beta_{i,j} + \varepsilon_{i,j,k}$, where $j = \{CgPHO4.CgPHO2, CgPHO4.pho2\Delta, pho4\Delta.CgPHO2, pho4\Delta.pho2\Delta\}$, k = 1,2 and $\varepsilon_{i,j,k}$ represents the noise. The three expression components can then be estimated as

$$X_{i,CgPHO4} = \beta_{i,CgPHO4.pho2\Delta} - \beta_{i,pho4.pho2\Delta}$$

$$X_{i,CgPHO2} = \beta_{i,pho4\Delta.CgPHO2} - \beta_{i,pho4\Delta.pho2\Delta}, \text{and}$$

$$CO_i = (\beta_{i,CgPHO4.CgPHO2} - \beta_{i,pho4\Delta.CgPHO2}) - (\beta_{i,CgPHO4.pho2\Delta} - \beta_{i,pho4\Delta.pho2\Delta})$$

We performed this analysis for the 79 genes that are directly bound and induced by CgPho4 in *C. glabrata* (*Figure 5—source data 2*). The genes were then hierarchically clustered using Ward's method (*Ward, 1963*; *Murtagh and Legendre, 2014*) based on the estimates of the three expression components.

## ChIP-seq analysis

We combined automatic peak calling with manual curation in order to identify binding sites for ScPho4 and CgPho4 in *S. cerevisiae*, and CgPho4 and CgPho2 binding sites in *C. glabrata*. First, we mapped the Bio-ChIP and ChIP-exo sequencing data to the respective reference genomes, as we did for RNA-seq data. Second, we used the program GEM (v2.6, RRID:SCR_005339) (*Guo et al., 2012*) to automatically call peaks for ScPho4 and CgPho4 ('–smooth 30 –fold 1.5') with mock sample as control. GEM iteratively performs peak calling and de novo motif discovery to improve both sensitivity and specificity (*Guo et al., 2012*) – for CgPho4, GEM identified the same E-box motif 'CACGTG' as recognized by ScPho4 (*Zhou et al., 2011*) in both *S. cerevisiae* and *C. glabrata*. For CgPho2, however, GEM performed poorly, identifying <10 significant peaks and no significantly enriched motifs. Instead, we used MACS2 (RRID:SCR_013291) (*Zhang et al., 2008*) to call peaks for CgPho2 in *C. glabrata* with '-s 50 –bw 200 -q 0.01 –keep-dup auto –slocal 1000 -B –verbose 4 m 2 100 –call-summits', and identified 640 binding peaks with a fold enrichment (FE) greater than three over the mock sample (380 peaks with FE > 4 and 1565 peaks with FE > 2). De novo motif discovery using RSAT (RRID:SCR_008560) Peak-motifs algorithm (*Thomas-Chollier et al., 20112012*) identified 'CACAGA' as the top motif, which appears in 126 of the 380 (33.16%) peaks that have FE > 4. It bears no similarity with the in vitro identified ScPho2 motif ('ATTA') (*Zhu et al., 2009*; *Zhao et al., 2009*). Nor did it match any known motifs of yeast transcription factors (RRID:SCR_006893) (*de Boer*

*and Hughes, 2012*). In subsequent analyses, we used CgPho2 ChIP peaks for categorizing CgPho4 binding sites into ones co-bound by CgPho2 or those bound by CgPho4 only.

After the automatic peak calling, we manually curated the peaks with visual help from a genome browser tool, MochiView (RRID:SCR_000259) (*Homann and Johnson, 2010*). We calculated a fold enrichment trace for each ChIP sample by comparing it to its matching input (or mock, in case of ChIP-exo), using MACS2 following https://github.com/taoliu/MACS/wiki/Build-Signal-Track. This trace file is converted to WIG format (http://genome.ucsc.edu/goldenPath/help/wiggle.html) and loaded into MochiView, along with the coordinates of the GEM identified ChIP peaks. We also imported previously published datasets for ScPho4 occupancy in *S. cerevisiae* with or without ScPho2 in no Pi conditions (*Zhou et al., 2011*). We then examined each GEM identified ChIP peak, using the height and peak shape to identify spurious binding peaks and false negative ones. The curated datasets are presented as source data accompanying *Figure 4* and *Figure 5*.

To estimate the ratio of ScPho4 and CgPho4 occupancy without vs with Pho2 in *S. cerevisiae* (*Figure 4B*), we combined the published ScPho4 binding occupancy in no phosphate conditions (*Zhou et al., 2011*) with our data for CgPho4 in the *pho80Δ* background. For the former, we downloaded the WIG files from the NCBI GEO database for series GSE29506, sample GSM730517 (Pho4_ChIP_NoPi) and GSM730527 (Pho4_ChIP_dPho2_NoPi), and scaled the raw library sizes so that the average coverage per base pair is 1. Then these were analyzed along with our CgPho4 ChIP enrichment in *pho80Δ*. For each of the ScPho4 (74) and CgPho4 (115) ChIP peaks, we recorded the maximum value of the normalized read counts for ScPho4 and CgPho4, with or without Pho2, and calculated the ratio between the two. Then the $-\log_2$ transformed values were plotted (*Figure 4B*). A two-sided Student's t-test was performed to assess the significance of the difference in the group means.

To investigate the nucleosome and Cbf1 occupancy in *S. cerevisiae* at sites either bound by CgPho4 only or by both ScPho4 and CgPho4, we used the published data from Table S2 of (*Zhou et al., 2011*). We first translated the coordinates in the table from sacCer1 into sacCer3 using the Galaxy (RRID:SCR_006281) tool (*Afgan et al., 2016*). We then identified all 'CACGTG' motifs overlapping with the ChIP peaks for ScPho4 or CgPho4 identified in this study using Genomics-Ranges package in R (*Lawrence et al., 2013*), and extracted the corresponding 'Nucleosomes.High. Pi' and 'Cbf1.Enrichment.High.Pi' values from the table. Finally, the occupancy or enrichment are plotted separately for sites bound by both ScPho4 and CgPho4, and sites bound by only CgPho4. A two-sided Student's t-test was performed to assess the significance of the difference in the group means.

## Identification of CgPho4 targets in *S. cerevisiae* and *C. glabrata*

To identify ScPho4 and CgPho4 induced genes in *S. cerevisiae* and CgPho4 induced genes in *C. glabrata*, we assigned each ChIP peak to the nearest gene(s) on both sides, provided that the ChIP peak is upstream (in the 5' UTR) of the gene, and queried the gene induction results from the RNA-seq experiments. ChIP binding peaks associated with Pho4-dependent gene induction are identified as the direct targets of ScPho4 or CgPho4. The identified target genes are presented in the source data accompanying *Figure 4* and *Figure 5*.

### Annotating Pho4 targets in *C. glabrata*

Annotation of putative Pho4 targets in *C. glabrata* was based largely on information of the orthologs in the well-annotated *S. cerevisiae* and *C. albicans* genomes. Gene Ontology enrichment analysis was performed using the 'GO term finder' tool, and mapping genes to GO Slim terms performed using the 'GO Slim mapper', both available on the CGD website (http://www.candidagenome.org/). All genomic features were used as the background set. For the enrichment analysis, a threshold of p<0.1 was used for selecting significant GO terms. We focused on the 'Biological Processes' to identify the functional groups as reported in the results.

## Acknowledgements

We thank Cecile Fairhead, Rodney Rothstein, Aviv Regev and Suzanne Noble for sharing yeast and bacteria strains. Cecile Fairhead shared Pho4 and Pho2 ortholog sequences in the newly sequenced *glabrata* clade species. Elmar Czeko helped with ChIP-exo. We thank Marty Kreitman, Sebastian

Maerkl, Ruth Franklin, Wenfeng Qian and his lab members, Christopher Chidley, Joseph Piechura, Xiaoyu Zheng, Alex Nyugen-Ba, Dennis Wykoff, Andrew Murray, Julia Koehler, Yong Zhang and members of the O'Shea lab for discussion and critical reading of the manuscript. Illumina sequencing for RNA-seq and ChIP-seq was performed at the Bauer Core Facility, and the computational analysis done on the Odyssey cluster, both supported by the FAS Division of Science at Harvard University. The Howard Hughes Medical Institute supported this work.

# Additional information

### Competing interests
Erin K O'Shea: Chief Scientific Officer and a Vice President at the Howard Hughes Medical Institute, one of the three founding funders of eLife. The other authors declare that no competing interests exist.

### Funding

| Funder | Author |
|---|---|
| Howard Hughes Medical Institute | Bin Z He<br>Xu Zhou<br>Erin K O'Shea |

The funders had no role in study design, data collection and interpretation, or the decision to submit the work for publication.

### Author contributions
Bin Z He, Conceptualization, Resources, Data curation, Software, Formal analysis, Validation, Investigation, Visualization, Methodology, Writing—original draft, Writing—review and editing; Xu Zhou, Conceptualization, Resources, Data curation, Software, Formal analysis, Investigation, Visualization, Methodology, Writing—review and editing; Erin K O'Shea, Conceptualization, Supervision, Funding acquisition, Project administration, Writing—review and editing

### Author ORCIDs
Bin Z He http://orcid.org/0000-0002-3072-6238
Xu Zhou http://orcid.org/0000-0002-1692-6823
Erin K O'Shea http://orcid.org/0000-0002-2649-1018

### Decision letter and Author response
Decision letter https://doi.org/10.7554/eLife.25157.sa1
Author response https://doi.org/10.7554/eLife.25157.sa2

# Additional files

### Data availability
The following dataset was generated:

| Author(s) | Year | Dataset title | Dataset URL | Database and Identifier |
|---|---|---|---|---|
| He BZ, Zhou X, O'Shea EK | 2017 | Evolution of Reduced Co-Activator Dependence Led to Target Expansion of a Starvation Response Pathway | https://www.ncbi.nlm.nih.gov/geo/query/acc.cgi?acc=GSE97801 | Publicly available at the NCBI Gene Expression Omnibus (accession no: GSE97801) |

The following previously published dataset was used:

| | | | | Database and |
|---|---|---|---|---|

| Author(s) | Year | Dataset title | Dataset URL | Identifier |
|---|---|---|---|---|
| Zhou X, O'Shea EK | 2011 | Integrated approaches reveal determinants of genome-wide binding and function of the transcription factor Pho4 | https://www.ncbi.nlm.nih.gov/geo/query/acc.cgi?acc=GSE29506 | Publicly available at the NCBI Gene Expression Omnibus (accession no: GSE29506) |

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
