## [Decision Letter]

Thank you for submitting your article "Evolution of Reduced Co-Activator Dependence Led to Target Expansion of a Starvation Response Pathway" for consideration by *eLife*. Your article has been reviewed by two peer reviewers, and the evaluation has been overseen by Naama Barkai as the Senior Editor and Reviewing Editor. The reviewers have opted to remain anonymous.

The reviewers have discussed the reviews with one another and the Reviewing Editor has drafted this decision to help you prepare a revised submission.

The paper presents a nice set of experiments describing functional divergence of a specific transcription factors activity (Pho4) is budding yeast, and show that this divergence results from differential dependence on its co-activator. Both reviewers appreciated the topic and your study. However, they also noted that, as written, the study appears very specific and is unlikely to appeal to a broad audience.

This can be addressed through modified writing, which will introduce, motivate and explain your study in a broader context, perhaps including some of the suggestions given by reviewer #2.

In addition, please address all other comments by the reviewers, as described below.

*Reviewer #1:*

Summary:

In this paper, He, Zhou and O'Shea investigate the evolution of Pho-pathway regulation by comparing Pho4 transcription factors from several different yeast species in the *S. cerevisiae* background. They use traditional assays (Figures 1, 2) as well as genomic approaches such as ChIP-seq and RNA-seq. Overall, the experimental approach seems solid and the claims and conclusion in the paper are generally very well supported by the data. Moreover, the statistical analysis of the genomics data seems solid as well.

The main conclusion is that losing a dependence on the transcription factor Pho2 allowed the transcription factor Pho4 to activate more genes and possibly also other stress response pathways in addition to the phosphate starvation pathway and this seems well supported by the data. However, although technically sound the authors could do more to make observations of general interest which is appropriate for a journal such as *eLife* with a wide readership. That is, it would be great if the authors could better explain what general implications losing the Pho2-dependence has and what readers, who do not work on phosphate, can learn from this?

*Reviewer #2:*

The authors examine the role of the transcription factors Pho4 and Pho2 in the yeast phosphate response pathways. In the model *Saccharomyces cerevisiae*, the activity of Pho4 is dependent on its co-activator Pho2. In the *Candida glabrata*clade, this dependency appears to be lost and Pho4 has an expended set of target genes, including genes that are not directly related to phosphate starvation. The authors conclude that the loss of dependency has allowed for the gain of new target genes by this transcriptional regulator. The study is well performed and the evidence is based on several types of experiments that support the underlying model, including the comparison of several species in parallel. The manuscript is well written and the flow is smooth and clear.

My major concern is the lack of larger perspective on evolution or cell biology that would make the paper appealing to the larger audience of *eLife*. Although the title is broad and suggests that the authors study a general phenomenon, the Abstract and the article are narrowed down on *S. cerevisiae* and the specific PHO pathway. Unless one is interested in the biology of yeast or of this pathway, the interest for the findings could be limited because it may be difficult to see how this relates to larger questions. I do not question the fact that the findings have broader impacts (and exciting) but I could appreciate the broader impact because of my scientific background, not because of the way the paper is presented. I believe this can be largely easily answered by first stating what we know about the evolution of regulatory pathways, what we do not know and why we need to address this lack of knowledge. Then, present the evolution of this pathway across yeast species as an opportunity to address it. For instance, how transcriptional regulators gain new target genes is a very important question, particularly in the view of the constraints that the authors mention in the Discussion (subsection “Evolutionary constraint and plasticity in the PHO response network”, third paragraph). Maybe the impact of the paper would be broadened if these questions were addressed in the Introduction. For instance, combinatorial TF regulation may actually allow for rapid rewiring of transcription programs by changing TF-TF interactions rather than changing many binding sites or the binding specificity of specific TFs.

Other comments:

Several experiments are performed in genetic backgrounds in which PHO80 is deleted but I could not find the justification for this nor if this had an impact on the results and their interpretation.

Subsection “In *C. glabrata* CgPho4 binds to DNA and activates gene expression largely independent of Pho2”, first paragraph: It could be useful to discuss the fact that the dependency does not need to be qualitative but could be quantitative. Regulatory networks are often described in textbooks in qualitative terms, i.e. this regulates that, but we know, and as shown here, that these relationships are not on or off but gradual.

Subsection “Pho4 direct targets in *C. glabrata* may function beyond phosphate homeostasis”: A great addition to the novelty of the paper and its impact on evolution would be to see whether the genes that are specifically regulated by Pho4 in the *C. glabrata*clade were "ready" to be Pho4 regulated. For instance, if these genes had already Pho4 binding sites before the loss of Pho2 dependency, this would show that the recruitment of new genes in a circuit proceeds first by the gain of TF binding sites and then by changes in trans-regulators. The authors discuss this possibility but it seems that a simple reconstruction of the history of the TF binding sites through bioinformatics analysis could be enough to test this model, at least partly.

Finally, it would be interesting to know how the dependency of TF evolves, is it by gain and losses of protein-protein interactions among TFs? It is known for instance that Pho4 and Pho2 interact with each other in *S. cerevisiae* – was this interaction lost in *C. glabrata*? This is not required for the paper to be complete but would be a great addition.

---

## [Author Response]

*[…] Reviewer #1:*

*[…] The main conclusion is that losing a dependence on the transcription factor Pho2 allowed the transcription factor Pho4 to activate more genes and possibly also other stress response pathways in addition to the phosphate starvation pathway and this seems well supported by the data. However, although technically sound the authors could do more to make observations of general interest which is appropriate for a journal such as eLife with a wide readership. That is, it would be great if the authors could better explain what general implications losing the Pho2-dependence has and what readers, who do not work on phosphate, can learn from this?*

*Reviewer #2:*

*[…] The manuscript is well written and the flow is smooth and clear.*

*My major concern is the lack of larger perspective on evolution or cell biology that would make the paper appealing to the larger audience of eLife. Although the title is broad and suggests that the authors study a general phenomenon, the Abstract and the article are narrowed down on S. cerevisiae and the specific PHO pathway. Unless one is interested in the biology of yeast or of this pathway, the interest for the findings could be limited because it may be difficult to see how this relates to larger questions. I do not question the fact that the findings have broader impacts (and exciting) but I could appreciate the broader impact because of my scientific background, not because of the way the paper is presented. I believe this can be largely easily answered by first stating what we know about the evolution of regulatory pathways, what we do not know and why we need to address this lack of knowledge. Then, present the evolution of this pathway across yeast species as an opportunity to address it. For instance, how transcriptional regulators gain new target genes is a very important question, particularly in the view of the constraints that the authors mention in the Discussion (subsection “Evolutionary constraint and plasticity in the PHO response network”, third paragraph). Maybe the impact of the paper would be broadened if these questions were addressed in the Introduction. For instance, combinatorial TF regulation may actually allow for rapid rewiring of transcription programs by changing TF-TF interactions rather than changing many binding sites or the binding specificity of specific TFs.*

We agree that placing our work in the broad context of regulatory evolution, and discussing the implications to other systems will greatly improve the appeal and impact of our work. To achieve this, we modified the Abstract and the beginning paragraph of the Introduction to place our work in the backdrop of gene regulatory evolution in general. After framing the importance of the problem and challenges, we focus on the central theme of our work – combinatorial regulation. We explained two shortcomings in the literature: few studies have analyzed evolutionary changes in combinatorial regulation and its effects on network function; and a strong bias in previous studies towards developmental networks. Our work targets these gaps by dissecting a known regulatory divergence involving combinatorial control in the phosphate starvation (PHO) response network in yeast. As starvation and stress response networks have very different architectures compared to developmental networks, our study will help identify shared and distinct evolutionary patterns across these two network types. We also significantly modified the first section of Discussion – instead of focusing on the PHO network itself, we compared our findings to existing knowledge of gene regulatory network evolution, pointing out similarities and differences from other types of GRNs. In particular, we focused on two topics – constraint and plasticity in GRN evolution – where we proposed that evolution of co-activator dependence provides a mechanism for rapid gain or loss of network targets compared to individual promoter evolution.

We hope that these revisions have improved our manuscript, helping it to reach a broad audience and resonate with researchers studying regulatory evolution in other systems.

*Other comments:*

*Several experiments are performed in genetic backgrounds in which PHO80 is deleted but I could not find the justification for this nor if this had an impact on the results and their interpretation.*

*PHO80* encodes the cyclin component of the cyclin-dependent-kinase (CDK) complex Pho80-Pho85, which negatively regulates Pho4 nuclear localization through phosphorylation (Kaffman et al. 1994; O’Neill et al. 1996). *pho80∆* thus genetically mimics the effect of phosphate starvation by causing Pho4 to be dephosphorylated and constitutively localized inside the nucleus in its functional form (Huang and O’Shea 2005).

The use of *pho80∆* instead of phosphate starvation serves two purposes in our study. First, for the phosphatase assay (Figure 2), it allowed us to decouple the activity of Pho4 orthologs from the viability of the strain carrying them. This means we can measure Pho4 ortholog activity even if the ortholog cannot activate the PHO response and therefore does not support growth on phosphate-limited medium. Second, because the central goal of our study is to dissect the evolution of Pho4 activity and how it influences network output, *pho80∆* allowed us to focus on the downstream targets of Pho4, ignoring starvation-induced but Pho4-independent genes.

To help explain the use of *pho80∆* in place of starvation, we described the role of phosphorylation in regulating Pho4 activity in the Introduction, added two sentences at the end of the first paragraph in the Results section, and explained the reason for using *pho80∆* in the second Results section.

We also verified that in the *pho80∆* background we can recapitulate the genes induced by ScPho4 under phosphate starvation (see Author response image 1). Four of the 28 genes previously identified as ScPho4 targets (Zhou and O’Shea 2011) are not included because of updated annotation in the latest genome release. Twenty of the 24 remaining genes are induced in *pho80∆* in a ScPho4-dependent manner. Three of the remaining four genes show more than two-fold induction by Pho4 in *pho80∆*, with *P*-values just over the threshold (0.05 after multi-test correction). We conclude that Pho4 targets identified in *pho80∆* closely recapitulate those identified under phosphate starvation.

**Author response image 1. respfig1:** Venn diagram showing overlap of genes induced by ScPho4 in the *pho80∆* vs. in phosphate starvation conditions (data from Zhou and O’Shea 2011).

*Subsection “In C. glabrata CgPho4 binds to DNA and activates gene expression largely independent of Pho2”, first paragraph: It could be useful to discuss the fact that the dependency does not need to be qualitative but could be quantitative. Regulatory networks are often described in textbooks in qualitative terms, i.e. this regulates that, but we know, and as shown here, that these relationships are not on or off but gradual.*

We fully agree with the reviewer on this. In fact, our phylogenetic survey using the quantitative phosphatase assay revealed substantial variation in the extent of Pho2 dependence among Pho4 orthologs from the *glabrata*clade species (Figure 2B). Treated as a quantitative trait, the extent of Pho2-dependence strongly correlates (negatively) with the number of genes induced in the *S. cerevisiae* background (Figure 3C, D). To emphasize the quantitative nature of the trait, we added a sentence at the end of the first Results section – “Notably, the extent of the reduction varies between the *glabrata*clade Pho4 orthologs, suggesting that the strength of combinatorial regulation is a quantitative trait that can be fine-tuned by mutations during evolution.” We also mentioned this in the Discussion section titled “Evolutionary Constraint and Plasticity in the PHO response network” – “However, it should be pointed out that the reduction in Pho2-dependence likely evolved gradually (Figure 2B, 3C), which means *cis* evolution could have accompanied the *trans* – while Pho4 expands its targets, promoter evolution could result in fixation of beneficial targets while removing spurious, non-beneficial ones.”

*Subsection “Pho4 direct targets in C. glabrata may function beyond phosphate homeostasis”: A great addition to the novelty of the paper and its impact on evolution would be to see whether the genes that are specifically regulated by Pho4 in the C. glabrata clade were "ready" to be Pho4 regulated. For instance, if these genes had already Pho4 binding sites before the loss of Pho2 dependency, this would show that the recruitment of new genes in a circuit proceeds first by the gain of TF binding sites and then by changes in trans-regulators. The authors discuss this possibility but it seems that a simple reconstruction of the history of the TF binding sites through bioinformatics analysis could be enough to test this model, at least partly.*

This is an intriguing question. To rephrase it, the question is “Does CgPho4 gain its new targets by exploiting cryptic DNA motifs not accessible to the ancestral Pho4?” We have considered this question and performed an analysis similar to what reviewer #2 suggested. Specifically, we took CgPho4 direct targets identified both in its own genome and in *S. cerevisiae*, and looked for gene orthology relationships. We found that the only CgPho4 targets that are orthologous with one another in the two species backgrounds were those involved in phosphate homeostasis, e.g. *PHO84*.

Does this mean all of CgPho4’s new targets in its own genome were de novo gains through promoter evolution? We think not. Then why can’t we identify *S. cerevisiae* genes that are orthologs of CgPho4 targets in *C. glabrata* and still retain the ability to be regulated by CgPho4? We think this is most likely because short DNA sequence motifs are rapidly gained and lost during evolution. Cryptic motifs – those that are inaccessible to the TF – are especially vulnerable to such a turnover process due to lack of purifying selection. As a result, the ancestral cryptic Pho4 motifs that were exploited by CgPho4 are almost certainly lost in the modern *S. cerevisiae* genome, explaining our observation of no overlap in this case.

In the text, we included a section at the end of the Results section to explain the logic above. “It is worth mentioning that, although CgPho4 induces more genes in *S. cerevisiae* as well as in *C. glabrata*, there is virtually no overlap between its targets in *S. cerevisiae* and in *C. glabrata*, except for those involved in phosphate homeostasis (as shown in Figure 6A).”

*Finally, it would be interesting to know how the dependency of TF evolves, is it by gain and losses of protein-protein interactions among TFs? It is known for instance that Pho4 and Pho2 interact with each other in S. cerevisiae – was this interaction lost in C. glabrata? This is not required for the paper to be complete but would be a great addition.*

We agree that this an outstanding question, and we are actively planning on resolving it. At the moment, although the detailed molecular mechanism of how CgPho4 became less Pho2-dependent is not clear, both our transcriptional and binding assay results are consistent with the hypothesis that CgPho4 evolved to be stronger in DNA-binding and in inducing gene expression, making it less dependent on Pho2 (Figure 4).

Regarding the interaction between CgPho4 and CgPho2, it is clearly not lost – our ChIP-exo data showed co-localization of CgPho2 to the majority of CgPho4 bound sites, and that in a small fraction of CgPho4 bound sites, its binding is weakened or lost when CgPho2 is deleted. In fact, loss or reduction of Pho4-Pho2 interaction is not likely to be behind the observed reduced dependence, because a mutant of ScPho4 unable to interact with ScPho2 has significantly decreased, rather than enhanced activity (Springer et al. 2003).

In the Discussion section titled “Evolutionary Constraint and Plasticity in the PHO response network”, we added a sentence explaining our favored hypothesis – “While the mechanism for this reduction in co-activator dependence is not clear, both transcriptional and binding assays are consistent with the hypothesis that CgPho4 evolved to be stronger in DNA-binding and in inducing gene expression (Figure 4).”